# On Three-Layer Data Markets

Alireza Fallah [*]    Michael I. Jordan[*]    Ali Makhdoumi [†]    Azarakhsh Malekian [‡]

## Abstract

We study a three-layer data market comprising users (data owners), platforms, and a data buyer. Each user benefits from platform services in exchange for data, incurring privacy loss when their data, albeit noisily, is shared with the buyer. The user chooses platforms to share data with, while platforms decide on data noise levels and pricing before selling to the buyer. The buyer selects platforms to purchase data from. We model these interactions via a multi-stage game, focusing on the subgame Nash equilibrium. We find that when the buyer places a high value on user data (and platforms can command high prices), all platforms offer services to the user who joins and shares data with every platform. Conversely, when the buyer's valuation of user data is low, only large platforms with low service costs can afford to serve users. In this scenario, users exclusively join and share data with these low-cost platforms. Interestingly, increased competition benefits the buyer, not the user: as the number of platforms increases, the user utility does not necessarily improve while the buyer utility improves. However, increasing the competition improves the overall utilitarian welfare. Building on our analysis, we then study regulations to improve the user utility. We discover that banning data sharing maximizes user utility only when all platforms are low-cost. In mixed markets of high- and low-cost platforms, users prefer a minimum noise mandate over a sharing ban. Imposing this mandate on high-cost platforms and banning data sharing for low-cost ones further enhances user utility.

## 1 Introduction and Overview of Results

In the digital era, online platforms have become integral to our daily lives, offering a plethora of services that range from social networking to personalized shopping experiences. While these platforms provide convenience and connectivity, they also engage in extensive data collection practices. Users' interactions with these platforms generate vast amounts of data, encompassing personal preferences, behaviors, and other sensitive information. This data, a valuable commodity, is often shared with third-party buyers.

With the advent of new functionalities by information technology companies such as Apple or Google that reduce or eliminate tracking via third-party cookies, there has been an increased reliance on *first-party* data, which is information gathered by a company from its own user base. For example, as elaborated in Cross [2023b], major retailers such as BestBuy and Walmart capitalize on this data through loyalty programs, customer purchase records, and subscriptions. They do this by selling the data to advertisers who aim to place targeted ads on the retailers' websites or across various online platforms. In a similar vein, Cross [2023a] notes that Mastercard, a leading payment technology firm, trades cardholder transaction information via third-party online data

---

[*]University of California, Berkeley

[†]Fuqua School of Business, Duke University

[‡]Rotman School of Management, University of Toronto.

marketplaces and its internal "Data and Services" division. This practice grants numerous entities access to extensive consumer data and insights. For instance, their Intelligent Targeting service allows businesses to leverage "Mastercard 360° data insights" to create and execute advertising campaigns that specifically target potential "high-value" customers.

In this complex digital ecosystem, users face the dilemma of benefiting from the services provided by these platforms while potentially compromising their privacy. Platforms, on the other hand, must balance the profitability derived from data sharing with the need to maintain user trust and satisfaction. This interplay between users' privacy concerns and the economic incentives of platforms raises important questions about the sustainability of current data market practices. Notable regulatory actions have been taken to improve user experiences. For instance, the General Data Protection Regulation (GDPR) and the California Consumer Privacy Act (CCPA) have been introduced to protect user privacy and to limit the amount of data shared with the platforms. Although, at first sight, this helps users, there have been several studies pointing to its detrimental impact both on platforms and users. This is because these regulations have led to a more concentrated market structure by limiting competition in data markets. This results from the regulation's restrictions on data sharing, which may slow down data-based innovations and services. For instance, studies have shown that GDPR has led to a significant reduction in the number of available apps in the market, with new app entries falling by half following the regulation's implementation (Janssen et al. [2022]). This decline in market entries indicates a potential loss of innovation as new and smaller players find it increasingly difficult to compete under these regulations. This effect is particularly detrimental to small companies that rely on these data synergies for innovation and market growth (see also Gal and Aviv [2020]).

In this paper, we aim to understand the user and platform strategies that arise when considering the collection and usage of user data. We also explore regulatory frameworks that could be implemented to enhance user utility while maintaining the benefits provided by online platforms. We adopt an analytical-modeling-based approach and develop a framework to examine the choices of both users and platforms, taking into account users' privacy concerns. We also explore regulations and policies that could alleviate these concerns. In our model, multiple platforms interact with a user who owns data and seeks the services of these platforms, as well as with a third-party buyer interested in purchasing the user's data from the platforms. Throughout our study, we consider MasterCard as an example of such a platform. The user represents an individual in a specific population segment with particular preferences, and the buyer is conceptualized as a third-party advertiser aiming to understand and leverage these user preferences for its own benefit.

The utility of each platform is determined by three factors: the gains obtained from using user data to provide services, potential payments received from buyers purchasing this data, and the costs incurred in providing services to the user. Let us highlight the cost of services, as it plays an important role in our analysis. It is particularly important to distinguish between *high-cost* and *low-cost* platforms. The low-cost platforms are those that do not necessarily need to sell data to stay in the market, as their gains from the service already cover their costs of operation. In our leading examples, low-cost platforms are exemplified by payment-technology companies or big retailers, where either their operational costs per user are low, or their main source of revenue is not derived from selling data. Conversely, for high-cost platforms, the gains from service provision are not enough to offset their operational costs. Hence, they require a minimum level of data selling to be incentivized to enter the market (i.e., to provide services to the user). Examples of these platforms typically include smaller firms or companies that heavily rely on data collection for advertising to ensure their continued operation.

We model the interaction between the players as a multi-stage game in which, in the first stage,

the platforms decide whether they want to provide services to the user and the privacy level they want to deliver to the data buyer (so that the user is incentivized to share). We model the privacy level that the platform provides as a *noise level* added to the user data. Adding a suitable level of noise is common practice in, for example, the analysis of census data Abowd [2018] and is theoretically justified in the literature on differential privacy Dwork et al. [2014].

In the second stage of the game, the user decides which platform to join. The user's utility is the service quality of the platform, proportional to the amount of user information revealed to the platform minus the user's *privacy loss*. The privacy loss refers to the amount of information all platforms reveal to the data buyer. In the third and fourth stages, the platforms determine the price to charge the data buyer, and the buyer decides from which platforms to purchase. The data buyer's utility is the user's leaked information from all platforms minus the payments made to acquire this information. Again, the data buyer can be viewed as a third-party advertiser in our example mentioned above who pays the platforms to obtain user data. They then use this data for potential gain, with their gain being proportional to the accuracy of the user data obtained.

**Equilibrium characterization and insights:** We adopt the subgame Nash equilibrium as our equilibrium concept and use backward induction to characterize it. Finding the equilibrium of this game is complex mainly because the platforms' choice of noise level not only directly impacts the user's utility but also changes the data buyer's decision, which indirectly impacts the user's utility. Also, the competition among platforms further complicates the dynamic of the game. For instance, a platform may still be incentivized to sell data at a noise level higher than needed for the user to be willing to share their data. At first glance, this may seem unreasonable as it reduces the quality of the data that they can sell, and hence, the data buyer would pay less for it. However, increasing the privacy could persuade the user to forgo other platforms that offer less stringent privacy guarantees and only use this platform's service. As a result, this can give a monopoly to the platform in terms of access to the user's data, and hence, the data buyer would end up paying more to them.

We start our analysis of equilibrium by focusing on the case of two platforms. This setting allows us to infer insights about equilibrium properties that are applicable to more general cases. Here, we identify up to three primary regimes, depending on the data buyer's valuation of user data acquisition, denoted by $\beta$ in our model. When $\beta$ falls below a certain threshold, neither of the two platforms enters the market due to insufficient payment to cover service costs. This regime ceases to exist when at least one platform is low-cost, meaning it can cover service costs solely through service provision without needing to sell data. Conversely, when $\beta$ is sufficiently high, both platforms will participate in the market. Interestingly, in this case, and when the user is privacy-conscious and requires a minimum non-zero level of noise for data sharing, the user utility is equal to zero at equilibrium. However, as the data buyer compensates each platform at the marginal value of the data, their utility remains positive. Finally, there exists an intermediate regime of $\beta$, where only the lower-cost platform engages in the market. Here, both the user and the data buyer utilities are zero at equilibrium, indicating that having both platforms in the market to compete benefits the data buyers but not the user.

We demonstrate that these findings extend to the general scenario with $K > 2$ platforms. Specifically, we establish that when $\beta$ exceeds a certain threshold, signifying the data buyer's willingness to pay more for the user data, all platforms will enter the market and provides services to the user. Conversely, when $\beta$ is below this threshold, only low-cost platforms will participate, while high-cost ones will abstain. Although competition does not favor the user, it results in positive utility for the data buyer. Moreover, the overall utilitarian welfare increases proportionally

with the number of participating platforms.

**Regulation and policy design:** Building on our equilibrium characterization, we then study regulations that aim to improve user utility. Here, our analysis provides three main insights. Firstly, we consider a ban on platforms sharing user data with the data buyers. This regulation only maximizes the user utility if all the platforms are low-cost. Specifically, under this regulation, users obtain the best of both worlds: all platforms (that have a low cost for providing services) enter the market, and the user benefits from the services of all of them while incurring zero privacy loss. This observation suggests that if we only had large firms whose costs of providing service per user are small, then the draconian regulation that involves banning data sharing would have been optimal. However, this is no longer the case if the market includes both high and low-cost platforms.

In this case, our second insight reveals that imposing a uniform minimum privacy mandate improves user utility, particularly when the value of the user data to the data buyer is significant enough for platforms to charge a high price for it. This means that all platforms need to perturb the user data by adding to it a noise with a high enough noise level. Here, our analysis identifies specific conditions where GDPR-type policies, which limit user data usage, can benefit users. Finally, our third insight suggests that, despite potential implementation challenges, a non-uniform minimum privacy mandate — banning data sharing for low-cost platforms while imposing a minimum privacy standard for high-cost platforms — further enhances user utility. This regulation entails prohibiting large, low-cost platforms from sharing user data, while permitting smaller, high-cost platforms to do so, albeit in a limited manner through privacy mechanisms. This approach aligns with our earlier discussion that GDPR-type regulations disproportionately harm small businesses. Therefore, a non-uniform regulation favoring small businesses is preferred not only by these businesses but also by the users.

## 1.1 Related literature

Our paper broadly relates to the emerging literature on data markets and online platform behavior. Earlier works on this topic include Acquisti et al. [2016], Posner and Weyl [2018], Ali et al. [2019], Jones and Tonetti [2020], and Dosis and Sand-Zantman [2022] which study the consequences of protecting and disclosing personal information about individuals. Additionally, Acemoglu et al. [2022], Bergemann et al. [2020], Ichihashi [2020b], and Fainmesser et al. [2022] study the privacy consequences of data externality (whereby a user's data reveals information about others). Acemoglu et al. [2023a] study user-optimal privacy-preserving mechanisms that platforms can adopt to balance privacy and learning and Acemoglu et al. [2023b] develop an experimentation game to study the implications of the platform's information advantage in product offering. Finally, Ichihashi [2021] studies data intermediaries, and Ichihashi [2020a] studies consumer privacy choices while online platforms adjust their privacy guarantees dynamically to incentivize more data sharing from consumers.

More closely related to ours are papers that study data sharing among users, platforms, and third-party data buyers and, in particular, the impacts of banning such data sharing. In this regard, Bimpikis et al. [2021] and Argenziano and Bonatti [2023] develop a two-round model of the interaction between a user and two competing platforms that can collect data on user preferences (and potentially use it for price discrimination). Bimpikis et al. [2021] show that banning data sharing can hurt the user when the platforms offer complementary products, and Argenziano and Bonatti [2023] show that banning data sharing can hurt the user if the benefits of knowing user preferences by the seller and personalized services are limited. The above papers consider the

interactions between a user and a platform that itself uses the user data for price discrimination and, therefore, can potentially hurt the user. In contrast, we study a different problem that relates to how platforms sell user data to other third-party data buyers. Therefore, in addition to the differences in modeling choices and analysis, we depart from the above papers by considering a three-layer data market that includes the interaction between the platforms and the data buyer and focusing on deriving insights on effective regulations.

Also related to our work is Ravichandran and Korula [2019] who empirically study the effect of the data-sharing ban on platforms' revenues and Madsen and Vellodi [2023] who study the effects of a complete ban on innovation (we refer to Pino [2022] for a comprehensive survey on the microeconomics of data). We depart from these papers by focusing on the impact of regulations (and in particular data-sharing ban) on user's utility. Our paper also relates to the literature that studies various forms of monetizing user data by platforms, such as selling cookies Bergemann and Bonatti [2015], efficient pricing for large data sets Abolhassani et al. [2017], dynamic sales Immorlica et al. [2021], Drakopoulos and Makhdoumi [2023], monetization while ensuring the data is replicable Falconer et al. [2023], and the design and price of information where a data buyer faces a decision problem under uncertainty and can augment their initial private information with supplemental data from a data seller Bergemann et al. [2018].

Finally, our paper relates to the growing literature on data acquisition and the design of mechanisms for incentivizing data sharing, including Ghosh and Roth [2011], Ligett and Roth [2012], Nissim et al. [2014], Cummings et al. [2015], Chen et al. [2018], Chen and Zheng [2019], Fallah et al. [2023], Cummings et al. [2023], Fallah et al. [2022], and Karimireddy et al. [2022]. We depart from this line of work as we consider a three-layer market where the privacy cost occurs due to selling data to a third-party buyer.

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

# Supplementary Material

The supplementary material proceeds as follows. In Section A, we describe our model and the interactions among the platforms, the data buyer, and the user. We also present some properties of the revealed information measure in our setting such as its monotonicity and submodularity. In Section B, we introduce our equilibrium concept and provide preliminary characterizations of it. In Section C, we focus on a setting with two platforms and provide a full characterization of the equilibrium. The focus on two platforms allows us to describe some of the nuances of the interactions in our model. In Section D, we establish that our main characterizations continue to hold in the general setting with any number of platforms. We also provide several insights and comparative statics, leading to our discussion of the regulations in Section E. Finally, Section F contains all the omitted proofs from the text.

## A  Model

We consider $K \in \mathbb{N}$ *platform*s that are interacting with a *user* and a *data buyer*. The user benefits from using the services of these platforms. In return, the platforms acquire data about the user's preferences and behavior. The platforms may subsequently sell this data to a third party, such as an advertiser, which we refer to as a data buyer (or, interchangeably, a buyer). The data buyer is interested in learning about the user's preferences, which can adversely impact the user's utility. This negative impact can be because of price discrimination, unfairly targeted advertising, manipulation, or intrinsic privacy losses. In the example of MasterCard described in the introduction, MasterCard is one of the platforms, and the data buyer is any other firm that purchases user data from MasterCard.

Formally, we represent the user's data by $\boldsymbol{\theta} \in \mathbb{R}^d$, which follows a Gaussian distribution with zero mean and identity variance, that is, $\boldsymbol{\theta} \sim \mathcal{N}(0, \mathbb{I}_d)$. This vector $\boldsymbol{\theta}$ can be interpreted as the user's feature vector. Each platform $i \in [K] \triangleq \{1, \dots, K\}$ offers a service whose quality depends on how accurately the platform can learn the match between the user's feature vector and the platform's characteristics, i.e.,

$$\boldsymbol{x}_i^\top \boldsymbol{\theta}.$$

Here, $\boldsymbol{x}_i \in \mathbb{R}^d$ is a known unit-norm vector symbolizing the characteristic vector of platform $i$.

The user decides whether to share their data with each of the $K$ platforms in exchange for their service. We let $a_i \in \{0, 1\}$ denote whether the user has shared data with platform $i$ and $\boldsymbol{a} \in \{0, 1\}^K$ **denote the user strategy profile**. Specifically, if the user opts to share their data with platform $i$, meaning $a_i = 1$, then platform $i$ receives a signal

$$s_i := \boldsymbol{x}_i^\top \boldsymbol{\theta} + \mathcal{N}(0, 1).$$

The introduced noise ensures a degree of privacy concerning the user's data.[1]

The data buyer aims to estimate a linear function of the user's data, $\boldsymbol{y}^\top \boldsymbol{\theta}$. Here, $\boldsymbol{y} \in \mathbb{R}^d$ with $\|\boldsymbol{y}\| = 1$ denotes the data buyer's known characteristic vector. If platform $i$ possesses the user's data (i.e., $a_i = 1$), they present the data buyer with an option to acquire a noisy version of their

---

[1]We set the noise variance to one to simplify the results. However, the analysis remains valid for any (potentially heterogeneous) level of noise.

data at a price $p_i$. Therefore, the **data buyer strategy profile is $b \in \{0,1\}^K$**, with $b_i = 1$ denoting their acceptance of the platform $i$'s offer. The data buyer then receives a signal

$$\tilde{s}_i := s_i + \mathcal{N}(0, \sigma_i^2)$$

for the price $p_i$, where $\sigma_i^2$ represents the variance of the noise that platform $i$ adds to the data they share with the data buyer.

Lastly, each platform $i \in [K]$ decides on the noise level to add to the user data before offering it to the data buyer, denoted by $\sigma_i \in \mathbb{R}_+$. We find it more convenient to work with the standard deviation of the noise $\sigma_i$ and assume, without loss of generality, it is positive rather than the variance $\sigma_i^2$. Each platform also decides whether to *enter* the market and provide services to the user, denoted by $e_i \in \{0,1\}$. Therefore, **the platforms' strategy profile is $(\boldsymbol{\sigma}, \boldsymbol{e}) \in \mathbb{R}_+^K \times \{0,1\}^K$**.

We set $\tilde{s}_i = \text{NA}$ when the data buyer does not obtain any information from platform $i$. This can occur either because the user has not shared their data with platform $i$ or because the data buyer declines the offer from platform $i$ or the platform has not provided the service to the user. In this case, $p_i$ is irrelevant, and we set it to zero.

Throughout the paper, we make the following assumption:

**Assumption 1.** *Let*

$$\gamma_i = \boldsymbol{x}_i^T \boldsymbol{y} \text{ for all } i \in [K]$$

*denote the correlation between the user $i$'s data and the data buyer. Then, we assume*

$$(\boldsymbol{x}_i - \gamma_i \boldsymbol{y})^T (\boldsymbol{x}_j - \gamma_j \boldsymbol{y}) = 0 \text{ for all } i \neq j \in [K].$$

Note that under this assumption, the data offered by different platforms are viewed as substitutes for each other from the data buyer's perspective. However, if this assumption does not hold, a complementary effect could also be observed. To illustrate this, suppose we have $K = 2$ platforms where $x_1$ is orthogonal to $y$, i.e., $\gamma_1 = 0$, and also $x_2 = \gamma_2 y + (1 - \gamma_2)x_1$. In this scenario, the data from platform 1 alone has no value to the data buyer, as $x_1^\top \theta$ provides no information about $\theta^\top y$. However, if the data buyer decides to purchase data from the second platform, then the data from the first platform could also become valuable. To understand why, observe that the data from the second platform is a convex combination of two components: $x_1^\top \theta$ and $y^\top \theta$. Therefore, if the data buyer acquires the data from the second platform, learning $x_1^\top \theta$ could assist in better distinguishing these two components. In other words, the data from the first platform is valuable only if accompanied by the data from the second platform. This scenario exemplifies the complementary effect we described earlier. Assumption 1 excludes this possibility, allowing us to primarily focus on the substitution effect. This assumption also implies that the correlation among data for different platforms is given by

$$\mathbb{E}[\boldsymbol{x}_i \boldsymbol{x}_j] = \gamma_i \gamma_j.$$

## A.1 Utility functions

To define the utility functions, we need to first quantify how much information is revealed by observing the signals. We do so by using the following definition, which captures the reduction in uncertainty following the observation of a set of signals.

**Definition 1** (Revealed information). *Consider a random variable $Z$ drawn from the distribution $\pi(.)$. Let $\pi_{\mathcal{S}}(.)$ represent the posterior distribution of $Z$ upon observing a set of signals $\mathcal{S}$. The revealed information about $Z$ by observing $\mathcal{S}$ is quantified as the reduction in the mean-squared error of $Z$, i.e.,*

$$\mathcal{I}(Z \mid \mathcal{S}) = \mathbb{E}\left[(Z - \mathbb{E}_{Z\sim\pi}[Z])^2\right] - \mathbb{E}\left[(Z - \mathbb{E}_{Z\sim\pi_{\mathcal{S}}}[Z])^2\right],$$

*where the outer expectations are over the randomness in $Z$ and the signals.*

Measuring the revealed information, or information gain, through the variance of the posterior distribution has been used widely in economics, statistics, and learning theory (see, for example, Breiman et al. [1984], Bergemann et al. [2020], and Acemoglu et al. [2022]).

Given this definition, the user's utility function is given by

$$\mathcal{U}_{\text{user}}(\boldsymbol{\sigma}, \boldsymbol{e}, \boldsymbol{a}, \boldsymbol{p}, \boldsymbol{b}) := \sum_{i=1}^{K} a_i e_i \mathcal{I}_i - \alpha \, \mathcal{I}(\boldsymbol{\sigma}, \boldsymbol{e}, \boldsymbol{a}, \boldsymbol{b}), \tag{1}$$

where the term

$$\mathcal{I}_i \triangleq \mathcal{I}(\boldsymbol{x}_i^{\top}\boldsymbol{\theta} \mid s_i)$$

is the user's revealed information to platform $i$ if the user shares her information (i.e., when $a_i = 1$) and the platform is offering service to the user (i.e., when $e_i = 1$). The term

$$\mathcal{I}(\boldsymbol{\sigma}, \boldsymbol{e}, \boldsymbol{a}, \boldsymbol{b}) \triangleq \mathcal{I}(\boldsymbol{y}^{\top}\boldsymbol{\theta} \mid \tilde{\boldsymbol{s}}),$$

is the user's revealed information to the data buyer. Here, $\tilde{\boldsymbol{s}}$ denotes the vector $(\tilde{s}_i)_{i=1}^{K}$, which is the revealed information to the data buyer. Notice that the user's utility does not directly depend on $\boldsymbol{p}$, but we retain this notation to highlight that through the data buyer's decision, the revealed information depends on $\boldsymbol{p}$. Also, notice that in finding the revealed information to the data buyer, all that matters is the set $\{\sigma_i : \text{s.t. } a_i = b_i = e_i = 1\}$. Therefore, we also find it convenient to define

$$\mathcal{I}(\boldsymbol{\sigma}_S) \text{ for } \boldsymbol{\sigma}_S = (\sigma_i : i \in S) \text{ for any } S \subseteq [K], \tag{2}$$

where we may drop the subindex $S$ whenever the indices of the elements of $\boldsymbol{\sigma}$ are clear from the context.

The first term in (1) captures the user's gain from the service offered by each platform. Notice that this term is present if the platform has entered the market (i.e., $e_i = 1$) and if the user shares with the platform (i.e., $a_i = 1$). The second term represents the user's privacy loss from the sharing of their data with the data buyer. Additionally, $\alpha$ indicates the user's relative importance on these two opposing terms.

For each $i \in [K]$, platform $i$'s utility is given by

$$\mathcal{U}_i(\boldsymbol{\sigma}, \boldsymbol{e}, \boldsymbol{a}, \boldsymbol{p}, \boldsymbol{b}) := \begin{cases} a_i \mathcal{I}_i + b_i p_i - c_i & e_i = 1 \\ 0 & e_i = 0. \end{cases} \tag{3}$$

When the platform has not entered the market (i.e., $e_i = 0$), the platform's utility is zero. When the platform has entered the market (i.e., $e_i = 1$), the second term $b_i p_i$ represents the platform's revenue from selling the user's data to the data buyer. The first term $a_i \mathcal{I}_i$, similar to the user's utility, reflects the quality of service the platform provides (if the user ends up utilizing their

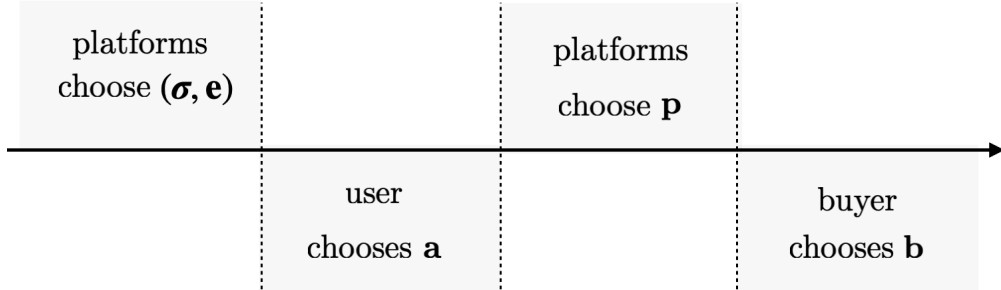

Figure 1: The timing of the decisions.

service). This term is concluded to capture the fact that a better service by the platform also enhances the platform's utility. For example, it could increase ad revenue or expand the user base. Lastly, the cost $c_i \in \mathbb{R}_+$ reflects the per-user cost incurred by the platform for providing the service.

Finally, the data buyer's utility is given by

$$\mathcal{U}_{\text{buyer}}(\boldsymbol{\sigma}, \boldsymbol{e}, \boldsymbol{a}, \boldsymbol{p}, \boldsymbol{b}) := \beta \, \mathcal{I}(\boldsymbol{\sigma}, \boldsymbol{e}, \boldsymbol{a}, \boldsymbol{b}) - \sum_{i=1}^{K} b_i p_i, \tag{4}$$

which is the data buyer's gain from learning the user's information minus the amount they pay to all the platforms to purchase their data. Here, $\beta \in [0,1]$ is some constant that captures the importance of the revealed information compared to the payments in the data buyer's utility.

We refer to the above game among the platforms, user, and buyer of data as a *three-layer data market* with parameters $(\alpha, \beta, \boldsymbol{c}, \boldsymbol{\gamma})$. Here is the timing of the game (see Figure 1):

1. In the first stage, all platforms simultaneously choose $(\boldsymbol{\sigma}, \boldsymbol{e}) \in \mathbb{R}_+^K \times \{0,1\}^K$ that specifies their noise levels and whether they want to enter the market to provide services to the user.

2. In the second stage, the user chooses $\boldsymbol{a} \in \{0,1\}^K$ that specifies which platforms to join and share data with.

3. In the third stage, the platforms simultaneously choose $\boldsymbol{p} \in \mathbb{R}^K$ that determines their offered prices.

4. In the fourth stage, the data buyer chooses $\boldsymbol{b} \in \{0,1\}^K$ that determines whether they accept or decline each offer made by the platforms.

## A.2 Revealed Information Properties

Before analyzing this game, we provide some properties of our notion of revealed information. In stating the properties, we recall that the leaked information to the data buyer depends on $\boldsymbol{e}$, $\boldsymbol{a}$, and $\boldsymbol{b}$ through the set of platforms that share user data with the data buyer and the noise level of each of them. Therefore, we make use of the shorthand notation for the leaked information given in (2). In what follows, we also consider the *lattice* $(\mathcal{L}, \leq)$, where

$$\mathcal{L} := \{\ell \mid \ell \in \{0,1\}^K\} \text{ and } \ell \leq \ell' \text{ if } \ell_i \leq \ell'_i \text{ for all } i \in [K]. \tag{5}$$

The revealed information admits the following closed-form expression.

**Lemma 1.** *For any $i \in [K]$, we have $\mathcal{I}_i = 1/2$. Moreover, for any $\boldsymbol{\sigma}$ and $S \subseteq [K]$ we have*

$$\mathcal{I}(\boldsymbol{\sigma}_S) = \mathbf{m}^T M^{-1} \mathbf{m},$$

*where $m_i = \gamma_i$, $[M]_{ii} = 2 + \sigma_i^2$, and $[M]_{ij} = \gamma_i \gamma_j$ for all $i, j \in S$ such that $i \neq j$.*

Lemma 1 follows by using the properties of multivariate normal distribution. We should highlight that for simplicity, we assumed that the user adds the same noise level to their data before sharing it with any platform, which results in $\mathcal{I}_i$ for all $I \in [K]$ being equal. Our analysis continues to hold for any heterogeneous level of noise.

Importantly, Lemma 1 enables us to obtain some structural properties of the revealed information, as we now demonstrate.

**Lemma 2** (monotonicity). *The revealed information is increasing in the set of platforms that share the user data with the data buyer and is decreasing in $\boldsymbol{\sigma}$, i.e., for any $\boldsymbol{\sigma}$ and $S \subseteq [K]$, $\mathcal{I}(\boldsymbol{\sigma}_S)$ is increasing in $S$ and decreasing in $\boldsymbol{\sigma}$.*

Lemma 2 simply states that more data sharing with the data buyer increases its revealed information. Moreover, data with a lower noise level increases the revealed information to the data buyer.

**Lemma 3** (submodularity in actions). *For a given vector of noise levels $\boldsymbol{\sigma}$, the revealed information is submodular in the set of platforms that share the user data with the data buyer, i.e., for any $\boldsymbol{\sigma}$, $S \subseteq T \subseteq [K]$, and $i \notin T$, we have*

$$\mathcal{I}((\sigma_i, \boldsymbol{\sigma}_S)) - \mathcal{I}(\boldsymbol{\sigma}_S) \geq \mathcal{I}((\sigma_i, \boldsymbol{\sigma}_T)) - \mathcal{I}(\boldsymbol{\sigma}_T).$$

Lemma 3 states the intuitively appealing result that as more data is shared with the data buyer, the marginal increase in the revealed information from one more unit of shared data becomes smaller.

**Lemma 4** (submodularity in noise variance). *For any $\boldsymbol{\sigma}$, $S \subseteq [K]$, $i \notin S$, and $j \in S$*

$$\mathcal{I}((\sigma_i, \boldsymbol{\sigma}_S)) - \mathcal{I}(\boldsymbol{\sigma}_S)$$

*is decreasing in $\sigma_i$ and increasing in $\sigma_j$.*

Lemma 4 states the data of platform $i$ reveals a relatively higher amount of information for smaller $\sigma_i$ values and for larger $\sigma_j$ values. To understand the former, let us compare the case of $\sigma_i \approx 0$ and $\sigma_i \to \infty$. When $\sigma_i \approx 0$, the data of platform $i$ reveals a lot, which helps the data buyer learn much better. However, when $\sigma_i \to \infty$, the data of platform $i$ does not contain any information, and therefore, the relative gain in the revealed information is $\approx 0$. To understand the former, let us again compare the case of $\sigma_j \approx 0$ and $\sigma_j \to \infty$. When $\sigma_j \approx 0$, the data of platform $j$ reveals a lot; therefore, the marginal increase in the revealed information of platform $i$ is small. However, when $\sigma_j \to \infty$, the data of platform $j$ does not contain any information, and therefore, the relative increase in the revealed information from platform $i$'s data is much larger.

# B    Preliminary Equilibrium Characterization

We adopt subgame perfect equilibrium as our solution concept, which means that the strategy profile of all players is a Nash equilibrium of every subgame of the game. We next formally define and characterize the subgame perfect equilibrium in our setting, starting from the last stage.

**Buyer equilibrium:** For a given $(\boldsymbol{\sigma}, e, \boldsymbol{a}, \boldsymbol{p})$, a data buyer profile $\boldsymbol{b}^{\mathrm{E}}$ is an equilibrium if it achieves the data buyer's highest utility, i.e.,

$$\boldsymbol{b}^{\mathrm{E}} \in \underset{\boldsymbol{b} \in \{0,1\}^K}{\arg \max} \mathcal{U}_{\text{buyer}}(\boldsymbol{\sigma}, e, \boldsymbol{a}, \boldsymbol{p}, \boldsymbol{b}).$$

Given that there is a finite set of $2^n$ possibilities, one of them achieves the maximum data buyer's utility, and therefore, an equilibrium buyer profile strategy always exists. We let $\mathcal{B}^{\mathrm{E}}(\boldsymbol{\sigma}, e, \boldsymbol{a}, \boldsymbol{p})$ be the set of such buyer equilibria.

**Price equilibrium:** For a given $(\boldsymbol{\sigma}, e, \boldsymbol{a})$, a pricing strategy by the platforms $\boldsymbol{p}^{\mathrm{E}}$ is an equilibrium if no platform has a profitable deviation (taking the data buyer's updated decision into account):

$$\mathcal{U}_i(\boldsymbol{\sigma}, e, \boldsymbol{a}, \boldsymbol{p}^{\mathrm{E}}, \boldsymbol{b}^{\mathrm{E}}) \geq \mathcal{U}_i(\boldsymbol{\sigma}, e, \boldsymbol{a}, (p_i, \boldsymbol{p}^{\mathrm{E}}_{-i}), \boldsymbol{b}) \text{ for all } i \in [K], \boldsymbol{b}^{\mathrm{E}} \in \mathcal{B}^{\mathrm{E}}(\boldsymbol{\sigma}, e, \boldsymbol{a}, \boldsymbol{p}^{\mathrm{E}}), p_i, \boldsymbol{b} \in \mathcal{B}^{\mathrm{E}}(\boldsymbol{\sigma}, e, \boldsymbol{a}, (p_i, \boldsymbol{p}^{\mathrm{E}}_{-i})).$$

Before proceeding with the rest of the game, we establish that the data buyer and price equilibria admit a simple characterization, stated next.

**Proposition 1.** *For a given $(\boldsymbol{\sigma}, e, \boldsymbol{a})$, there exists a unique price equilibrium given by*

$$p_i^* = \beta \left( \mathcal{I}(\boldsymbol{\sigma}, e, \boldsymbol{a}, \mathbf{1}) - \mathcal{I}(\boldsymbol{\sigma}, e, \boldsymbol{a}, (b_i = 0, \mathbf{1}_{-i})) \right) \text{ for all } i \in [K]$$

*and the corresponding unique buyer equilibrium is $b_i = 1$ for all $i \in [K]$, where $\mathbf{1}_{-i}$ is the vector of all ones for all $j \in [K] \setminus \{i\}$.*

The proof of Proposition 1, given in the appendix, directly follows from the submodularity of the revealed information established in Lemma 3 for the following reason. Consider platform $i \in [K]$ and suppose the contrary that in equilibrium, the buyer is not purchasing from it. This platform has a profitable deviation by lowering its price so that the data buyer finds it optimal to purchase from it. Also, notice that the submodularity of the revealed information implies that if one more platform's data is shared with the data buyer, the marginal increase in the revealed information decreases. Therefore, if the data buyer was purchasing from another platform $j \neq i$, they still find it optimal to purchase from that platform.

Proposition 1 characterizes the equilibrium of the third and fourth stages of the game. Therefore, it remains to characterize the user sharing profile and the platform's noise level and entry decisions in equilibrium, taking into account the equilibrium of the subsequent stages that we have already identified.

**User equilibrium:** For a given $(\boldsymbol{\sigma}, e)$, a user sharing profile $\mathbf{a}^{\mathrm{E}}$ is an equilibrium if it achieves the user's highest utility, i.e.,

$$\boldsymbol{a}^{\mathrm{E}} \in \underset{\boldsymbol{a} \in \{0,1\}^K}{\arg \max} \mathcal{U}_{\text{user}}(\boldsymbol{\sigma}, e, \boldsymbol{a}, \boldsymbol{p}^*, \mathbf{1}).$$

Again, given that there are $2^n$ possibilities, one of them achieves the maximum user's utility, and therefore, an equilibrium sharing profile strategy always exists. We let $\mathcal{A}^{\mathrm{E}}(\boldsymbol{\sigma}, e)$ be the set of such user equilibria. This set has the following important lattice structure that we use in the rest of the analysis.

**Lemma 5.** *For a given $(\boldsymbol{\sigma}, e)$, the set of user equilibria $\mathcal{A}^{\mathrm{E}}(\boldsymbol{\sigma}, e)$ is a sublattice of the lattice $(\mathcal{L}, \leq)$ (defined in (5)). Therefore, it has a maximum and a minimum.*

For a platform strategy profile $(\boldsymbol{\sigma}, \boldsymbol{e})$, we select the platforms' Pareto-optimal one if there are multiple user equilibria. Using Lemma 5, such a user equilibrium exists and is the one that shares with the highest number of platforms.[2]

**Platform equilibrium:**   A noise level and entry strategy $(\boldsymbol{\sigma}^{\mathrm{E}}, \boldsymbol{e}^{\mathrm{E}})$ is an equilibrium if no platform has a profitable deviation, i.e.,

$$\mathcal{U}_i(\boldsymbol{\sigma}^{\mathrm{E}}, \boldsymbol{e}^{\mathrm{E}}, \boldsymbol{a}^{\mathrm{E}}, \boldsymbol{p}^*, \mathbf{1}) \geq \mathcal{U}_i((\sigma_i, \boldsymbol{\sigma}^{\mathrm{E}}_{-i}), (e_i, \boldsymbol{e}^{\mathrm{E}}_{-i}), \boldsymbol{a}, \boldsymbol{p}^*, \mathbf{1})$$

for all $i \in [K], \boldsymbol{a}^{\mathrm{E}} \in \mathcal{A}^{\mathrm{E}}(\boldsymbol{\sigma}^{\mathrm{E}}, \boldsymbol{e}^{\mathrm{E}}), \boldsymbol{a} \in \mathcal{A}^{\mathrm{E}}((\sigma_i, \boldsymbol{\sigma}^{\mathrm{E}}_{-i}), (e_i, \boldsymbol{e}^{\mathrm{E}}_{-i}))$.

We refer to a platform's noise level and entry decision as the platform's decision and to the corresponding equilibrium as the platforms' equilibrium. The following characterizes the set of possible platforms' equilibria.

**Proposition 2.** *For any platforms' equilibrium strategy* $(\boldsymbol{\sigma}, \boldsymbol{e})$*, we must have*

$$\mathcal{A}^{\mathrm{E}}(\boldsymbol{\sigma}, \boldsymbol{e}) = \{(a_i = 1 \ : \ \text{for all } i \text{ s.t. } e_i = 1))\},$$

*that is, the user must find it optimal to share with all platforms that have entered the market.*

This proposition follows by noting that a platform that has entered the market can always increase its noise level so that the user finds it optimal to share with the platform.

Motivated by Proposition 2 and Lemma 5, we conclude that for any set of platforms that have entered the market (captured by $\boldsymbol{e} \in \{0,1\}^K$), the set of possible noise levels in equilibrium is $\Sigma_{\mathbf{1},\boldsymbol{e}}$, where we have

$$\Sigma_{\mathbf{a},\boldsymbol{e}} := \{(\sigma_i \ : \ i \text{ s.t. } e_i = 1) \ : \ \boldsymbol{a} \text{ is the highest element of } \mathcal{A}^{\mathrm{E}}(\boldsymbol{\sigma}, \boldsymbol{e})\} \text{ for all } \boldsymbol{a} \in \{0,1\}^K. \quad (6)$$

We are interested in characterizing $\Sigma_{\mathbf{1},\boldsymbol{e}}$ for any $\boldsymbol{e}$ and finding the equilibrium that belongs to one of these sets for an equilibrium choice of $\boldsymbol{e}$.

Throughout the paper, we focus on the case in which the user asks for some level of privacy in order to share their data. This means that if the noise levels are all zero, then the user does not share with the platforms. More formally, we make the following assumption:

**Assumption 2.** *The weight of privacy on the user utility, i.e.,* $\alpha$*, is large enough such that, for any* $\boldsymbol{e} \in \{0,1\}^K$*, the point* $(\sigma_i = 0 \ : \ i \text{ s.t. } e_i = 1)$ *belongs to the set* $\Sigma_{\mathbf{0},\boldsymbol{e}}$.

We will further clarify this assumption's role in the next section.

## C   Equilibrium Characterization with $K = 2$ Platforms

We begin our analysis by considering a setting with $K = 2$ platforms. We then show that the main insights from this analysis extend to the general setting with $K > 2$ platforms.

Depending on which platforms enter the market, we have four cases:

---

[2]We can view this as Stackelberg Nash equilibrium in which because the platforms move first, we break the ties in their favor. Similar concepts have appeared in the literature with different terms, such as sender-preferred equilibrium in Bayesian persuasion [Kamenica and Gentzkow, 2011].

**Case 1 where both platforms enter the market:** In this case, the user's utility if they share their data with both platforms is given by[3]

$$\mathcal{U}_{\text{user}}(\sigma_1, \sigma_2) := 1 - \alpha \mathcal{I}(\sigma_1, \sigma_2), \tag{7}$$

and the user's utility if they share their data with platform $i \in \{1, 2\}$ only is equal to

$$\mathcal{U}_{\text{user}}(\sigma_i) := \frac{1}{2} - \alpha \mathcal{I}(\sigma_i). \tag{8}$$

Recall the definition of $\Sigma_{a,e}$ with $e = (1,1)$ in (6). To simplify the notation, we drop the $e$ and use $\tilde{\Sigma}_a$. In particular,

$$\tilde{\Sigma}_{(1,1)} := \{(\sigma_1, \sigma_2) : \mathcal{U}_{\text{user}}(\sigma_1, \sigma_2) = \max\left(\mathcal{U}_{\text{user}}(\sigma_1, \sigma_2), \mathcal{U}_{\text{user}}(\sigma_1), \mathcal{U}_{\text{user}}(\sigma_2), 0\right)\}.$$

Similarly, $\tilde{\Sigma}_{(1,0)}$ and $\tilde{\Sigma}_{(0,1)}$ denote the cases where $\mathcal{U}_{\text{user}}(\sigma_1)$ and $\mathcal{U}_{\text{user}}(\sigma_2)$, respectively, have the highest value. Finally,

$$\tilde{\Sigma}_{(0,0)} := \{(\sigma_1, \sigma_2) : 0 = \max\left(\mathcal{U}_{\text{user}}(\sigma_1, \sigma_2), \mathcal{U}_{\text{user}}(\sigma_1), \mathcal{U}_{\text{user}}(\sigma_2), 0\right)\}.$$

Notice that we operate under Assumption 2, which implies that $\tilde{\Sigma}_{(0,0)}$ is non-empty. In particular, for $K = 2$, this assumption translates to

$$\alpha > \frac{4 - \gamma_1^2 \gamma_2^2}{2\left(\gamma_1^2 + \gamma_2^2 - \gamma_1^2 \gamma_2^2\right)}. \tag{9}$$

Figure 2a depicts the sets $\tilde{\Sigma}_{(1,1)}$, $\tilde{\Sigma}_{(0,1)}$, $\tilde{\Sigma}_{(1,0)}$, and $\tilde{\Sigma}_{(0,0)}$ as a function of the noise variance that the two platforms add to the data, i.e., $(\sigma_1^2, \sigma_2^2)$, for parameters $\gamma_1 = 0.8$ and $\gamma_2 = 0.7$.

Using Proposition 2, the noise level equilibrium is in the set $\tilde{\Sigma}_{(1,1)}$. In fact, we argue that it must belong to $\tilde{\Sigma}_{(1,1)} \cap \tilde{\Sigma}_{(0,0)}$ because, otherwise, one of the platforms can increase its utility by decreasing its noise level such that the user still shares with it.

Now, for any $(\sigma_1, \sigma_2) \in \tilde{\Sigma}_{(1,1)} \cap \tilde{\Sigma}_{(0,0)}$, the only possible deviation for platform 1 is to increase $\sigma_1$ to $\tilde{\sigma}_1$ to move to the point $(\tilde{\sigma}_1, \sigma_2)$ which resides in $\tilde{\Sigma}_{(1,0)}$, exactly next to the boundary with $\tilde{\Sigma}_{(1,1)}$ (see Figure 2a). At first glance, it might look like this can never be a deviation because it reduces the revealed information to the buyer. However, notice that, by this action, platform 1 pushes platform 2 out of the market because $(\tilde{\sigma}_1, \sigma_2)$ is in the yellow region $\tilde{\Sigma}_{(1,0)}$ in which the user does not share their data with the second platform. Therefore, it is possible that the first platform's payment increases, as they are now the only platform with access to the data. Note that this is the only deviation that we need to consider. Because if it ends up not being a deviation, it would imply that a further increase of $\sigma_1$ into the yellow region is also not profitable. Also, it is evident that increasing $\sigma_1$ but staying in the $\tilde{\Sigma}_{(1,1)}$ region or decreasing it and going into the $\tilde{\Sigma}_{(0,0)}$ region in which the user shares no data are not profitable deviations. Considering these deviations results in the following.

**Lemma 6.** *Suppose Assumptions 1 and 2 hold. Assuming both platforms have entered the market, there exists an equilibrium noise level $(\sigma_1, \sigma_2) \in \tilde{\Sigma}_{(1,1)} \cap \tilde{\Sigma}_{(0,0)}$ if and only if $\alpha \geq \bar{\alpha} \approx 1.884$. In particular, for $\alpha \geq \bar{\alpha}$, $(\sigma_1, \sigma_2)$ is an equilibrium where*

$$\frac{2 + \sigma_1^2}{\gamma_1^2} = \frac{2 + \sigma_2^2}{\gamma_2^2} = 2\alpha - 1. \tag{10}$$

---

[3]With a slight abuse of notation, we drop the other dependencies of the user utility here.

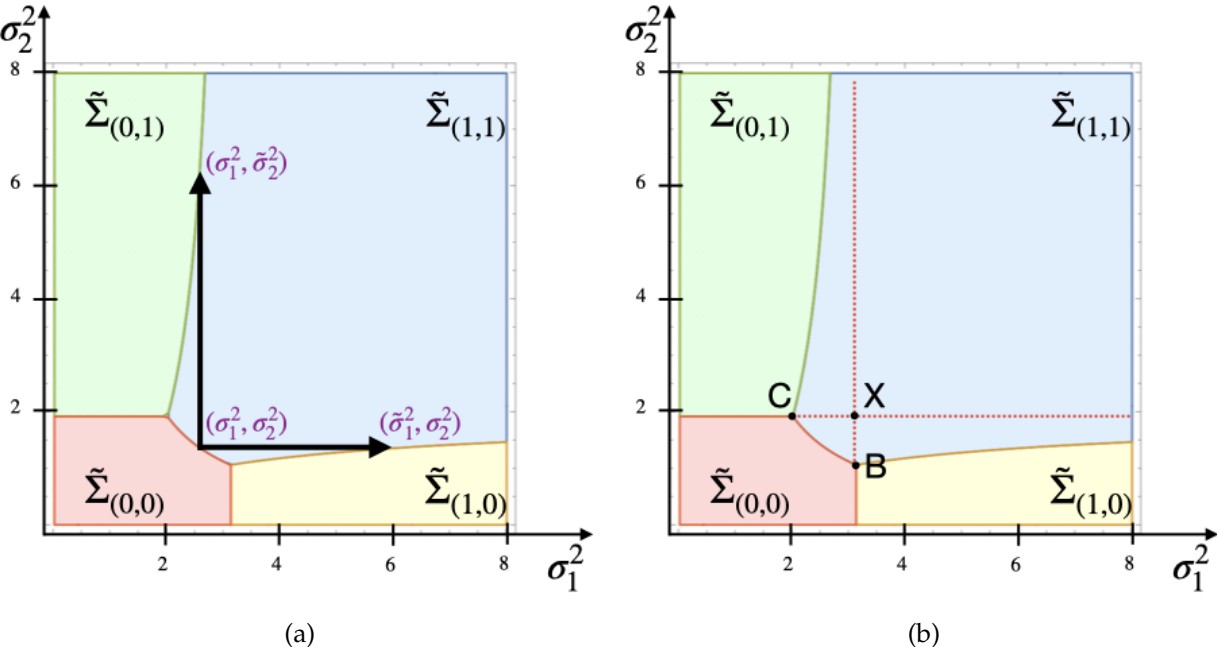

Figure 2: Four possible sets $\tilde{\Sigma}_a$ for $a \in \{0,1\}^2$ for a setting with $K = 2$ platforms, $\gamma_1 = .8$, and $\gamma_2 = .7$.

Lemma 6 is under the assumption that both platforms enter the market. The final point to check is whether any platform has an incentive to enter the market.

**Proposition 3.** *Suppose Assumptions 1 and 2 hold. Also, recall $\bar{\alpha} \approx 1.88$ from Lemma 6. If an equilibrium exists in which both platforms enter the market, then*

$$\alpha \geq \bar{\alpha} \quad \text{and} \quad \beta \geq 2\alpha \left(\max\{c_1, c_2\} - \frac{1}{2}\right). \tag{11}$$

*Conversely, for any $\alpha \geq \bar{\alpha}$ and any $\beta \geq \bar{\beta}(\alpha)$ such an equilibrium in which both platforms enter the market exists, where*

$$\bar{\beta}(\alpha) \triangleq \left(2 + \frac{1}{\alpha - 1}\right) \alpha \left(\max\{c_1, c_2\} - \frac{1}{2}\right). \tag{12}$$

Both parts of the result provide a lower bound on $\beta$, with the difference that the first part establishes a necessary condition for an equilibrium to exist, while the second part presents a sufficient condition. Note that the ratio between these two lower bounds converges to one as $\alpha$ grows. However, we next highlight a subtle difference between the two results. Note that, for any value $\xi$ greater than $2\left(\max c_1, c_2 - \frac{1}{2}\right)$, we can choose sufficiently large $\alpha$ and $\beta$ such that an equilibrium exists for these parameters and the ratio of $\beta$ over $\alpha$ is less than or equal to $\xi$. In other words, we have

$$\inf\left\{\frac{\beta}{\alpha} \;\middle|\; \alpha, \beta \text{ s.t. an equilibrium exists}\right\} = 2\left(\max\{c_1, c_2\} - \frac{1}{2}\right).$$

This, in fact, is a direct consequence of the second part of Proposition 3 along with the fact that $\bar{\beta}$ converges to $2\left(\max\{c_1, c_2\} - \frac{1}{2}\right)$ as $\alpha$ grows.

However, the infimum mentioned above does not imply that any $\alpha$ and $\beta$ satisfying the condition (11) will necessarily result in an equilibrium. In fact, as our proof establishes, to have an equilibrium for $\alpha = \bar{\alpha}$, we must have $\beta \geq \bar{\beta}(\bar{\alpha})$. In other words, we have

$$\sup_{\alpha \geq \bar{\alpha}} \inf \left\{ \frac{\beta}{\alpha} \mid \beta \text{ s.t. an equilibrium exists with the given } \alpha \right\} = \bar{\beta}(\bar{\alpha}).$$

**Cases 2 and 3 where only one platform enters the market:** Suppose only platform 1 enters the market at some equilibrium. In this case, the first platform wants to choose the noise level $\sigma_1$ as small as possible but also large enough so that user shares their data. Therefore, the platform chooses $\sigma_1$ at the value that makes the user indifferent between sharing and not sharing (recall that we assumed the user breaks the ties in favor of the platforms), i.e.,

$$\mathcal{U}_{\text{user}}(\sigma_1) = \frac{1}{2} - \alpha \mathcal{I}(\sigma_1) = 0, \tag{13}$$

which implies $\sigma_1^2 = 2\alpha\gamma_1^2 - 2$. Notice that this is equivalent to the $\sigma_1$ corresponding to points $B$ and $X$ in Figure 2b.

Now, for this noise level to be an equilibrium, we need to verify two potential deviations: (i) platform one has no incentive to abstain from participating, i.e., their utility should be nonnegative, and (ii) the second platform has no profitable deviation by entering the market. Using (13), the first condition simply implies

$$\frac{1}{2} - c_1 + \beta \frac{1}{2\alpha} \geq 0. \tag{14}$$

For the second deviation check, we argue that it suffices to ensure the second platform's utility at point $B$ in Figure 2b is nonnegative.

Let us denote the noise levels corresponding to point $B$ as $(\sigma_1^B, \sigma_2^B)$. To understand why we only need to check the point $B$, consider that, given platform one's chosen noise level (equal to $\sigma_1^B$), if the second platform enters the market and selects a certain noise variance, it will result in a point on the line $BX$. The smallest noise level for the second platform that motivates the user to share their data with them (and thus has the potential to be profitable) is $\sigma_2^B$. One might argue that the second platform could opt for a very large noise level so that the vector of noise levels falls into the green region $\tilde{\Sigma}_{(0,1)}$, implying that only the second platform receives the data. However, this is not feasible, as the line $BX$ stays within the blue region $\tilde{\Sigma}_{(1,1)}$ for any value of $\sigma_2$ (and in fact, it becomes tangent to the border of $\tilde{\Sigma}_{(0,1)}$ and $\tilde{\Sigma}_{(1,1)}$ as $\sigma_2$ approaches infinity). To confirm this, we need to demonstrate that

$$\mathcal{U}_{\text{user}}(\sigma_1^B, \sigma_2) = 1 - \alpha \mathcal{I}(\sigma_1^B, \sigma_2) \geq \frac{1}{2} - \alpha \mathcal{I}(\sigma_2) = \mathcal{U}_{\text{user}}(\sigma_2),$$

holds for any $\sigma_2$. This is indeed the case, given that due to the submodularity of the revealed information, $\mathcal{I}(\sigma_1^B, \sigma_2) - \mathcal{I}(\sigma_2)$ decreases with increasing $\sigma_2$ and equals $\mathcal{I}(\sigma_1^B) = \frac{1}{2\alpha}$ when $\sigma_2 = \infty$.

Now, checking the second platform's utility at point $B$ along with (14) yields the following result.

**Proposition 4.** *Suppose Assumptions 1 and 2 hold. For any $i \in \{1, 2\}$, there exists an equilibrium in which only platform $i$ enters the market if and only if $c_i \leq c_{-i}$ and*

$$\beta \in \left[ 2\alpha(c_i - \frac{1}{2}), 2\alpha(c_{-i} - \frac{1}{2}) \right].$$

**Case 4 where no platform enters the market:** Finally, we ask whether there is an equilibrium in which no platform enters the market. In such a scenario, we need to ensure no platform has a profitable deviation by entering the market and choosing some noise level. For instance, if platform 1 wants to deviate, the smallest noise level that they would choose would be $\sigma_1^B$ as it makes the user indifferent between sharing and not sharing their data with them. In this case, the platform's utility, as derived in (14), is given by

$$\frac{1}{2} - c_1 + \beta \frac{1}{\alpha},$$

which is nonpositive when $\beta \leq 2\alpha(c_1 - \frac{1}{2})$. Making the same argument for the second platform's deviation, we obtain the following result.

**Proposition 5.** *Suppose Assumptions 1 and 2 hold. There exists an equilibrium in which neither of the platforms enters the market if and only if*

$$\beta \leq 2\alpha \left( \min\{c_1, c_2\} - \frac{1}{2} \right).$$

## C.1 Insights from the case with $K = 2$

We conclude this section by highlighting a few implications and insights of our results for the case $K = 2$, which, as we will see, also extend to the general case. Note that the term $\beta$ determines the value the buyer places on the user's data. The higher $\beta$ is, the more money the platform can earn by selling the user's data.

Our results indicate that there are, at most, three main regimes for $\beta$, and the market equilibrium depending on the interval $\beta$ falls into. First, as established in Proposition 5, when $\beta$ is small, the platform's monetary gain from selling data is limited, so platforms may decide not to provide any service as they cannot fully compensate for their service costs. This regime would not exist if at least one of the platforms operates at a *low cost*, meaning its gain from the user's data already compensates for the cost of service (this corresponds to the case $c_i \leq 1/2$ for some $i$).

On the other hand, as shown in Proposition 3, when $\beta$ is large enough and platforms can charge a high price for the user's data, both platforms would enter the market. Finally, there is an intermediary regime for $\beta$ where only the platform with the lower cost enters the market, and the platform with a higher service cost opts out.

Interestingly, the user's utility in all these cases would be equal to zero. However, the buyer's utility could be positive when both platforms enter the market. As we establish next, these observations continue to hold in the general case with $K > 2$ platforms.

# D  Equilibrium Characterization with $K > 2$ Platforms

This section shows how our equilibrium characterization extends to a general setting with any $K > 2$ number of platforms. In what follows, we make use of the following definition.

**Definition 2** (High- and low-cost platforms). *We say a platform $i \in [K]$ is of high cost if $c_i > \frac{1}{2}$ and is of low cost if $c_i < \frac{1}{2}$.*

The comparison of the cost to $1/2$ in the above definition comes from the fact that the leaked information to any platform is $\mathcal{I}_i = 1/2$ and therefore, high-cost platforms enter the market only if the payment they receive from the buyer is large enough. Conversely, low-cost platforms enter the

market irrespective of the payment they receive from the buyer. We can view low-cost platforms as large platforms with small operation costs that do not need extra payments from the data buyer to enter the market and provide services. High-cost platforms, however, are small platforms with large operation costs that do need extra payments from the buyer to be able to provide service.

**Theorem 1.** *Suppose Assumptions 1 and 2 hold. There exists $\bar{\alpha}$, $\bar{\beta}$, and $\underline{\beta}$ such that for $\alpha \geq \bar{\alpha}$, we have the following:*

- *If $\beta \geq \bar{\beta}$, an equilibrium of the game exists. Moreover, in any equilibrium, all platforms enter the market, the user utility is zero, and the buyer purchases the data of all platforms and has a strictly positive utility.*

- *If $\beta \leq \underline{\beta}$, an equilibirum of the game exists. Moreover, in any equilibrium, only low-cost platforms enter the market, the user utility is zero, and the buyer purchases the data of low-cost platforms and has a strictly positive utility only if there is more than one low-cost platform.*

Let us explain the qualifiers of Theorem 1 and then explain its intuition. First, notice that similar to the setting with $K = 2$ platforms, for small enough $\alpha$ and also for intermediary values of $\beta$, a pure-strategy equilibrium may not exist. Therefore, we focus on large enough $\alpha$ and also avoid intermediary values of $\beta$. In the proof of Theorem 1, we have provided explicit expressions for $\bar{\alpha}$, $\bar{\beta}$, and $\underline{\beta}$.

To gain the intuition of the first part of Theorem 1, notice that for large enough $\beta$ the data buyer gains a lot when the user's data is revealed to them and therefore is willing to pay a high price for the user data. This, in turn, means that even high-cost platforms find it optimal to enter the market. As established in Proposition 1 and Proposition 2, in equilibrium, the user shares with all platforms, and the data buyer purchases from all platforms. To see that the user utility becomes zero in equilibrium, suppose the contrary, that the user utility is strictly positive. This means that the equilibrium noise level strategy is in the interior of the set $\Sigma_{1,1}$. Given that noise level strategy is in the interior of $\Sigma_{1,1}$, there exists $i \in [K]$ such that by decreasing its noise level, we remain in the set $\Sigma_{1,1}$ (i.e., the user still finds it optimal to share with all platforms), but this deviation increases the payment and therefore the utility of platform $i$. Also, Lemma 3 guarantees that the user still finds it optimal to share with all platforms.

To gain the intuition of the second part of Theorem 1, notice that for small enough $\beta$, the data buyer's gain when the user's data is revealed to them is small and therefore, the buyer is willing to pay only a low price for the user data. This means that high-cost platforms will never find it optimal to enter the market. Nonetheless, the low-cost platforms always enter the market. The fact that the user shares with all the low-cost platforms (that have entered), the buyer purchases from them, and the user utility becomes zero in equilibrium follows from a similar argument to the one discussed above.

Theorem 1 leads to the following insights.

**Competition among platforms helps the buyer and not the user:** Let us first characterize the equilibrium when we have only one platform.

**Proposition 6.** *Suppose Assumption 1 holds. When $K = 1$, we have the following:*

- *Suppose $\alpha \leq 1/\gamma_1^2$. If $c_1 \leq 1/2 + \beta\gamma_1^2/2$ (which holds whenever $c_1 \leq 1/2$), then in equilibrium, the platform enters the market and chooses $\sigma_1 = 0$, the user shares with the platform, and the buyer purchases the data. In this case, the user utility is $(1 - \alpha\gamma_1^2)/2$, and the buyer utility is zero. Otherwise, if $c_1 > 1/2 + \beta\gamma_1^2/2$, then the platform does not enter the market, and all utilities are zero.*

- *Suppose $\alpha > 1/\gamma_1^2$. If $c_1 \leq 1/2 + \beta/2\alpha$, then in equilibrium, the platform enters the market and chooses $\sigma_1 = \sqrt{2(\alpha\gamma_1^2 - 1)}$, the user shares with the platform and the buyer purchases the data. In this case, both the user utility and the buyer utility are zero. Otherwise, if $c_1 > 1/2 + \beta/2\alpha$, then the platform does not enter the market, and all utilities are zero.*

We are interested in the second regime where $\alpha$ is large enough so that the user is not incentivized to share with the platform, irrespective of the noise level choice of the platform. In this case, as established in Proposition 6, both the user and the data buyer utilities are zero. Now, let us compare this setting to a setting with $K \geq 2$ platforms that are competing to obtain the user's data and sell it to the data buyer, established in Theorem 1. Interestingly, we observe that competition does not help the user (as the user obtains zero utility for any $K$). However, increasing the number of platforms from $K = 1$ to $K \geq 2$ improves the data buyer's utility. To gain the intuition for this observation, notice that the user's utility will always be zero because otherwise, the platforms can always decrease their noise levels by a small margin so that the user still shares and increases their payments. Now, why does the competition help the data buyer? This is because of the data externality among the platforms' data about the user, similar to that of Acemoglu et al. [2022] and Bergemann et al. [2020]. In particular, the data of a platform (because of the submodularity of the revealed information, proved in Lemma 3) decreases the marginal value of another platform's data, and therefore, the platforms end up competing with each other, and sell their data at lower marginal prices.

**More platforms imply higher utilitarian welfare:** Let us first find the overall utilitarian welfare in equilibrium, where utilitarian welfare is defined as the sum of the utilities of the platforms, the user, and the data buyer.

**Corollary 1.** *Consider $\bar{\alpha}$, $\bar{\beta}$, and $\underline{\beta}$ established in Theorem 1. For $\alpha \geq \bar{\alpha}$, we have the following:*

- *If $\beta \geq \bar{\beta}$, the utilitarian welfare in equilibrium becomes*

$$K \left( \frac{1}{2} + \frac{\beta}{2\alpha} \right) - \sum_{i=1}^{K} c_i.$$

- *If $\beta \leq \underline{\beta}$, the utilitarian welfare in equilibrium becomes*

$$(\textit{number of low-cost platforms}) \left( \frac{1}{2} + \frac{\beta}{2\alpha} \right) - \sum_{i \in [K]\,:\,c_i < \frac{1}{2}} c_i.$$

Notably, Corollary 1 shows that, within the ranges of each case, the overall utilitarian welfare is increasing in $\beta$ and decreasing in $\alpha$. This is because the payments always cancel each other out, and as $\beta$ increases, the data buyer's gain becomes larger, and as $\alpha$ increases, the user's loss becomes larger. Moreover, in the first case, the utilitarian welfare is increasing in the number of platforms, and in the second case, it is increasing in the number of low-cost platforms.

# E   Regulations

In this section, we consider regulations of the three-layer data market to improve the user utility. As we have established in Theorem 1, the user utility in equilibrium is zero. We now explore

whether we can improve the user utility by imposing a regulation. As discussed in the introduction, one possible regulation is to impose a limit on the amount of information each platform can leak about the user to the data buyer. This means imposing a lower bound on the noise level. Let us formally define this regulation.

**Definition 3** (Minimum privacy mandate). *A minimum privacy mandate, represented by $\bar{\boldsymbol{\sigma}} = (\bar{\sigma}_1, \ldots, \bar{\sigma}_K)$, is a regulation that mandates that each platform $i \in [K]$ must choose a noise level above $\bar{\sigma}_i \in \mathbb{R}_+$. A special case of this regulation is uniform minimum privacy mandate is such that $\bar{\sigma}_i = \bar{\sigma}$ for all $i \in [K]$. Another special case of this regulation is to ban the platforms from sharing with the buyer, i.e., $\bar{\sigma}_i = \infty$ for all $i \in [K]$.*

**Uniform minimum privacy mandate versus ban:** One may conjecture that banning the platforms from sharing their data with the buyer is user-optimal in the class of all uniform minimum privacy mandate regulations because it reduces the privacy loss of the user to zero. We next prove that this is not necessarily true. The main intuition is that if we ban data sharing, then only low-cost platforms enter the market. This, in turn, implies that the user does not obtain the gain from the services provided by the high-cost platforms, and this loss in the service gain can dominate the gain in privacy loss. We next formalize this intuition.

**Proposition 7.** *Consider $\alpha \geq \bar{\alpha}$ for $\bar{\alpha}$ established in Theorem 1. We have:*

- *If all platforms are low-cost, then irrespective of the minimum privacy mandate, all platforms enter the market in equilibrium. In this case, banning data sharing achieves a higher user utility than any (uniform or non-uniform) minimum privacy mandate.*

- *Otherwise, if there is at least one high-cost platform, then there exists $\tilde{\beta}$, $\underline{\beta}$, and $\bar{\sigma}$ such that*

    - *If $\beta \geq \tilde{\beta}$, then under uniform minimum privacy mandate $\bar{\sigma}$ all platforms enter the market, and the user utility is higher than banning data sharing.*
    - *If $\beta \leq \underline{\beta}$, then irrespective of the minimum privacy mandate, only low-cost platforms enter the market. In this case, banning data sharing achieves a higher user utility than any uniform minimum privacy mandate.*

The first part is rather straightforward: if all platforms are low-cost, then they all enter the market and provide services to the user. Banning data sharing is user-optimal because it decreases the privacy loss of the user to zero while the user obtains the service gains from all platforms. The second part is more nuanced: here, if $\beta$ is large enough, then all platforms enter the market. Now, there are two opposing forces in place. If we ban data sharing, only low-cost platforms enter the market, and user utility then comprises only the service gained from low-cost platforms, while the privacy loss becomes zero. Therefore, banning data sharing decreases the service gains but also decreases the privacy losses. If we do not ban data sharing, the user utility comprises the service gains from all platforms and the privacy loss incurred by the data sharing of all platforms with the buyer. Therefore, not banning data sharing increases the service gains but also increases the privacy losses. For a large enough minimum uniform privacy mandate, and when there is at least one high-cost platform, the service gains of the user utility dominate the privacy loss, and therefore, not banning data sharing becomes the user-optimal regulation. Finally, to understand the last part, we observe that for small enough $\beta$, only low-cost platforms enter the market, and therefore, the situation is effectively the same as the first part of Proposition 7. Notice that the assumption of large enough $\alpha$ and large/small enough $\beta$ is adopted for the same reason as that of Theorem 1 so that an equilibrium exists.

**Non-uniform versus uniform minimum privacy mandate:** As noted above, when all platforms are low-cost or when $\beta$ is small enough so that high-cost platforms do not enter the market, then banning data sharing is optimal in the class of all (uniform or non-uniform) minimum privacy mandates. However, corresponding to the second part of Proposition 7, the user prefers a minimum privacy mandate to banning data sharing when there is at least one high-cost platform, and $\beta$ is large enough. This is because if we ban data sharing, the high-cost platforms do not enter, while with the "right" choice of uniform minimum privacy mandate, they enter the market. In this case, the user benefits from the services, while the privacy loss that they incur is small. However, with a uniform minimum privacy mandate, the low-cost platforms (that enter the market regardless of the minimum privacy mandate in place) also bring about privacy loss to the user. Therefore, a natural idea is to have a non-uniform minimum privacy mandate to help the user further. We next formalize this idea.

**Proposition 8.** *Suppose there is at least one high-cost platform and consider $\alpha \geq \bar{\alpha}$ for $\bar{\alpha}$ established in Theorem 1. There exists $\hat{\beta}$ and $\bar{\sigma}$ such that for $\beta \geq \hat{\beta}$ banning the low-cost platforms from data sharing and imposing minimum privacy mandate equal to $\bar{\sigma}$ for all high-cost platforms generates a higher user utility than any non-uniform minimum privacy mandate.*

With the non-uniform minimum privacy mandate described above, the user obtains the service gain of the low-cost platforms and incurs zero privacy loss from them. Moreover, for large enough $\beta$, the high-cost platforms all enter the market. Therefore, the user gains from the high-cost platforms' services, and when $\bar{\sigma}$ is large enough, it incurs a small privacy loss. Notice that, as established in Proposition 7, with the optimal uniform minimum privacy mandate, both the low-cost and high-cost platforms enter the market. We prove that if we impose the same minimum privacy mandate but only for high-cost platforms and ban the low-cost platforms, the user obtains the same service gain from all platforms but incurs a smaller privacy loss. Moreover, all platforms still find it optimal to enter the market.

Proposition 7 and Proposition 8 establish that banning the data market is not necessarily user-optimal, especially when we have high-cost platforms (i.e., smaller platforms whose cost of providing the service is high). Moreover, a uniform minimum privacy mandate improves the user utility because it also incentivizes the high-cost platforms to enter the market to provide service to the user. Finally, a non-uniform minimum privacy mandate that bans the low-cost (i.e., large) platforms from data sharing and imposes a minimum privacy mandate on high-cost (i.e., small) platforms further improves the user utility. This is because this regulation not only incentivizes the high-cost platforms to enter the market to provide service to the user but also zeros out the privacy loss of the low-cost platforms.

## F Proofs

This section included the omitted proofs from the text.

**Proof of Lemma 1**

Without loss of generality, we prove the claim for $S = [K]$. Let us recall a property of normal distributions: If $\mathbf{x}$ is a $n$-dimensional normal random variable is partitioned as

$$\mathbf{x} = \begin{bmatrix} \mathbf{x}_1 \\ \mathbf{x}_2 \end{bmatrix} \text{ with sizes } \begin{bmatrix} k \times 1 \\ (n-k) \times 1 \end{bmatrix}$$

with with sizes

$$\begin{bmatrix} k \times k & k \times (n-k) \\ (n-k) \times k & (n-k) \times (n-k) \end{bmatrix}$$

and accordingly $\boldsymbol{\mu}$ is partitioned as

$$\boldsymbol{\mu} = \begin{bmatrix} \boldsymbol{\mu}_1 \\ \boldsymbol{\mu}_2 \end{bmatrix}$$

with sizes

$$\begin{bmatrix} k \times 1 \\ (n-k) \times 1 \end{bmatrix}$$

and $\boldsymbol{\Sigma}$ is partitioned as

$$\boldsymbol{\Sigma} = \begin{bmatrix} \boldsymbol{\Sigma}_{11} & \boldsymbol{\Sigma}_{12} \\ \boldsymbol{\Sigma}_{21} & \boldsymbol{\Sigma}_{22} \end{bmatrix}$$

with sizes

$$\begin{bmatrix} k \times k & k \times (n-k) \\ (n-k) \times k & (n-k) \times (n-k) \end{bmatrix}$$

then the distribution of $\mathbf{x}_1$ conditional on $\mathbf{x}_2 = \mathbf{a}$ is normal with covariance matrix

$$\boldsymbol{\Sigma}_{11} - \boldsymbol{\Sigma}_{12}\boldsymbol{\Sigma}_{22}^{-1}\boldsymbol{\Sigma}_{21}.$$

The proof of this lemma follows directly from the above fact. ∎

**Proof Lemma 2**

Both parts follow from the conditional expectation being the MSE estimator. To see the first part, notice that we can always throw away data points. To see the second part, notice that we can always add extra noise to the data points. ∎

**Proof of Lemma 3**

Using Lemma 1, we first derive the leaked information to the buyer. In particular, if both platforms enter the market and the user's data is shared with them and then sold to the buyer, then the leaked information is given by

$$\mathcal{I}(\sigma_1, \sigma_2) = \frac{\gamma_1^2(2 + \sigma_2^2) + \gamma_2^2(2 + \sigma_1^2) - 2\gamma_1^2\gamma_2^2}{(2 + \sigma_1^2)(2 + \sigma_2^2) - \gamma_1^2\gamma_2^2}.$$

On the other hand, if the user's data is only shared with the buyer through platform $i$, then the leaked information to the buyer is given by

$$\mathcal{I}(\sigma_i) = \frac{\gamma_i^2}{2 + \sigma_i^2}.$$

To prove this lemma, without loss of generality, we need to establish that if the buyer has the extra data of platform $j$, the gain in acquiring the data of platform $i$ is smaller. By invoking the fact

mentioned in the proof of Lemma 1 for the normal variable $(\boldsymbol{y}^T\boldsymbol{\theta}, \boldsymbol{x}_i^T\boldsymbol{\theta} + \mathcal{N}(0,1) + \mathcal{N}(0,\sigma_i^2), \boldsymbol{x}_j^T\boldsymbol{\theta} + \mathcal{N}(0,1) + \mathcal{N}(0,\sigma_j^2))$ conditional on the rest of the data points, the inequality stated in the lemma becomes equivalent to

$$\frac{\tilde{\gamma}_i^2}{2+\sigma_i^2} \geq \frac{\tilde{\gamma}_i^2(2+\sigma_j^2) + \tilde{\gamma}_j^2(2+\sigma_i^2) - 2\tilde{\gamma}_i^2\tilde{\gamma}_j^2}{(2+\sigma_i^2)(2+\sigma_j^2) - \tilde{\gamma}_i^2\tilde{\gamma}_j^2} - \frac{\tilde{\gamma}_j^2}{2+\sigma_j^2},$$

where for all $\ell \in [K]$, $\tilde{\gamma}_\ell = \gamma_\ell(\tilde{\text{var}})^{1/2}$ where $\tilde{\text{var}}$ is the variance of $\boldsymbol{\theta}^T\boldsymbol{\theta}$ conditional on the rest of the data points. The above inequality holds true as it is equivalent to

$$\tilde{\gamma}_i^2\tilde{\gamma}_j^2\left(\frac{\tilde{\gamma}_i^2}{2+\sigma_i^2} + \frac{\tilde{\gamma}_j^2}{2+\sigma_i^2}\right) \leq 2\tilde{\gamma}_i^2\tilde{\gamma}_j^2$$

that holds as $\tilde{\gamma}_i, \tilde{\gamma}_j \leq 1$. ∎

## Proof of Lemma 4

We prove part one and part two follows an identical argument. We use a similar argument to the one presented in the proof of Lemma 3. We establish this inequality by considering a one-by-one change in the noise variance.

If $\boldsymbol{\sigma}_{-j} = \boldsymbol{\sigma}'_{-j}$ and ${\sigma'_j}^2 \geq \sigma_j^2$, then the inequality becomes equivalent to

$$\frac{\tilde{\gamma}_i^2(2+\sigma_j^2) + \tilde{\gamma}_j^2(2+\sigma_i^2) - 2\tilde{\gamma}_i^2\tilde{\gamma}_j^2}{(2+\sigma_i^2)(2+\sigma_j^2) - \tilde{\gamma}_i^2\tilde{\gamma}_j^2} - \frac{\tilde{\gamma}_j^2}{2+\sigma_j^2}$$

$$\leq \frac{\tilde{\gamma}_i^2(2+{\sigma'_j}^2) + \tilde{\gamma}_j^2(2+\sigma_i^2) - 2\tilde{\gamma}_i^2\tilde{\gamma}_j^2}{(2+\sigma_i^2)(2+{\sigma'_j}^2) - \tilde{\gamma}_i^2\tilde{\gamma}_j^2} - \frac{\tilde{\gamma}_j^2}{2+{\sigma'_j}^2}.$$

The above inequality holds because the function

$$x \mapsto \frac{\tilde{\gamma}_i^2(2+x) + \tilde{\gamma}_j^2(2+\sigma_i^2) - 2\tilde{\gamma}_i^2\tilde{\gamma}_j^2}{(2+\sigma_i^2)(2+x) - \tilde{\gamma}_i^2\tilde{\gamma}_j^2} - \frac{\tilde{\gamma}_j^2}{2+x}$$

is increasing, which can be seen by evaluating its derivative. ∎

## Proof of Proposition 1

We first argue that, for any $i \in [K]$, the equilibrium price of platform $i$ must be such that the buyer finds it optimal to purchase from this platform in the corresponding buyer equilibrium. To see this notice that if the buyer does not purchase from platform $i$, the platform's utility from interacting with the buyer becomes 0. Now, if the platform deviates and selects the price

$$p_i = \beta\left(\mathcal{I}(\boldsymbol{\sigma}, \boldsymbol{e}, \boldsymbol{a}, \boldsymbol{1}) - \mathcal{I}(\boldsymbol{\sigma}, \boldsymbol{e}, \boldsymbol{a}, (b_i = 0, \boldsymbol{1}_{-i})))\right)$$

then the buyer always finds it optimal to purchase from this platform, and therefore, the platform's utility weakly increases from 0. This is because for any $\boldsymbol{b}_{-i}$, we have

$\mathcal{U}_{\text{buyer}}(\boldsymbol{\sigma}, \boldsymbol{e}, \boldsymbol{a}, (p_i, \boldsymbol{p}_{-i}), (b_i = 1, \boldsymbol{b}_{-i})) - \mathcal{U}_{\text{buyer}}(\boldsymbol{\sigma}, \boldsymbol{e}, \boldsymbol{a}, (p_i, \boldsymbol{p}_{-i}), (b_i = 0, \boldsymbol{b}_{-i}))$

$= \beta\left[(\mathcal{I}(\boldsymbol{\sigma}, \boldsymbol{e}, \boldsymbol{a}, (b_i = 1, \boldsymbol{b}_{-i})) - \mathcal{I}(\boldsymbol{\sigma}, \boldsymbol{e}, \boldsymbol{a}, (b_i = 0, \boldsymbol{b}_{-i}))) - (\mathcal{I}(\boldsymbol{\sigma}, \boldsymbol{e}, \boldsymbol{a}, \boldsymbol{1}) - \mathcal{I}(\boldsymbol{\sigma}, \boldsymbol{e}, \boldsymbol{a}, (b_i = 0, \boldsymbol{1}_{-i})))\right]$

$\overset{(a)}{\geq} 0$

where (a) follows from Lemma 3. ∎

**Proof of Lemma 5**

This lemma directly follows the submodularity of the leaked information in actions, established in Lemma 3. ∎

**Proof of Proposition 2**

Suppose the contrary that there exists a platform equilibrium strategy $(\boldsymbol{\sigma}, \boldsymbol{e})$ for which $e_i = 1$, $a_i = 0$ and $\boldsymbol{a} \in \mathcal{A}^{\mathrm{E}}(\boldsymbol{\sigma}, \boldsymbol{e})$. We argue that platform $i$ has a profitable deviation. In particular, notice that the current utility of platform $i$ is $-c_i$, as the buyer will not pay any non-zero price for this platform's data. Now, if platform $i$ chooses $\tilde{e}_i = 1$, $\tilde{p}_i = 0$ and $\tilde{\sigma}_i = \infty$, then in any $\boldsymbol{a} \in \mathcal{A}^{\mathrm{E}}((\tilde{\sigma}_i, \boldsymbol{\sigma}_{-i}), \boldsymbol{e})$ we must have that $a_i = 1$ and therefore the platform $i$'s utility increases to $\mathcal{I}_i - c_i$. ∎

**Proof of Lemma 6**

Recall that we argued that the equilibrium, if exists, belong to $\tilde{\Sigma}_{(1,1)} \cap \tilde{\Sigma}_{(0,0)}$. Furthermore, we stated that we only need to check deviations in Figure 2a. To make sure such a deviation does not happen for some equilibrium $(\sigma_1, \sigma_2) \in \tilde{\Sigma}_{(1,1)} \cap \tilde{\Sigma}_{(0,0)}$, we must have

$$\mathcal{U}_1((\sigma_1, \sigma_2)), \mathbf{1}, \mathbf{p}^*, \mathbf{1}) = \frac{1}{2} + \beta\left(\mathcal{I}(\sigma_1, \sigma_2) - \mathcal{I}(\sigma_2)\right) \geq \frac{1}{2} + \beta\mathcal{I}(\tilde{\sigma}_1) = \mathcal{U}_1((\tilde{\sigma}_1, \sigma_2)), (1, 0)), \mathbf{p}^*, \mathbf{1}). \tag{15}$$

Notice that $(\sigma_1, \sigma_2)$ is on the boundary of $\tilde{\Sigma}_{(1,1)}$ and $\tilde{\Sigma}_{(0,0)}$ which implies that

$$\mathcal{U}_{\mathrm{user}}(\sigma_1, \sigma_2) = 1 - \alpha\mathcal{I}(\sigma_1, \sigma_2) = 0, \tag{16}$$

and thus, $\mathcal{I}(\sigma_1, \sigma_2) = 1/\alpha$. Using this, we can rewrite the condition (15) as

$$\frac{1}{\alpha} \geq \mathcal{I}(\sigma_2) + \mathcal{I}(\tilde{\sigma}_1). \tag{17}$$

Notice that we can similarly consider the deviation of the second platform by increasing its noise variance to $\tilde{\sigma}_2^2$ in Figure 2a. To ensure this deviation is also not profitable, we should have

$$\frac{1}{\alpha} \geq \mathcal{I}(\sigma_1) + \mathcal{I}(\tilde{\sigma}_2). \tag{18}$$

The following lemma establishes the necessary and sufficient condition for (17) and (18) to hold for some $(\sigma_1, \sigma_2) \in \tilde{\Sigma}_{(1,1)} \cap \tilde{\Sigma}_{(0,0)}$.

Notice that if we move up on the boundary $\tilde{\Sigma}_{(1,1)} \cap \tilde{\Sigma}_{(0,0)}$, $\sigma_2$ and $\tilde{\sigma}_1$ both would increase, and hence, $\mathcal{I}(\sigma_2)$ and $\mathcal{I}(\tilde{\sigma}_1)$ both would decrease. Therefore, if (17) holds for some $(\sigma_1, \sigma_2) \in \tilde{\Sigma}_{(1,1)} \cap \tilde{\Sigma}_{(0,0)}$, then it would hold for any upper point on the boundary. Similarly, if (18) holds for some $(\sigma_1, \sigma_2) \in \tilde{\Sigma}_{(1,1)} \cap \tilde{\Sigma}_{(0,0)}$, then it would hold for any lower point on the boundary. Hence, if a point satisfies both conditions, then for any other point on the boundary $\tilde{\Sigma}_{(1,1)} \cap \tilde{\Sigma}_{(0,0)}$, at least one of the conditions would hold.

Next, we show that, for the point given by (10), both conditions hold if $\alpha \geq \bar{\alpha}$ and neither hold if $\alpha < \bar{\alpha}$. Given our discussion above, this would show that for $\alpha < \bar{\alpha}$, there is no point for which both conditions hold and hence the proof would be complete.

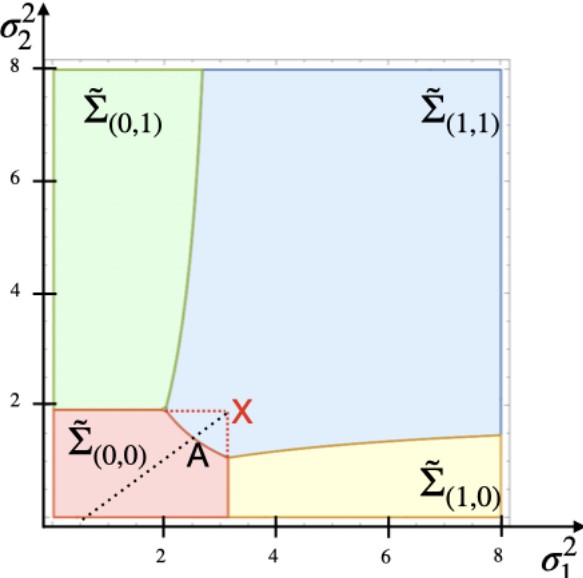

Figure 3: Illustrating the point given by (10).

Now, let us first see why the point given by (10) is actually on the boundary $\tilde{\Sigma}_{(1,1)} \cap \tilde{\Sigma}_{(0,0)}$. Consider point $X$ in Figure 3. For this point, we have

$$\frac{\gamma_1^2}{2 + \sigma_1^2} = \frac{\gamma_2^2}{2 + \sigma_2^2} = \frac{1}{2\alpha}.$$

Hence, the line

$$\frac{2 + \sigma_1^2}{\gamma_1^2} = \frac{2 + \sigma_2^2}{\gamma_2^2} \tag{19}$$

passes through point $X$, and therefore, it intersects with the boundary $\tilde{\Sigma}_{(1,1)} \cap \tilde{\Sigma}_{(0,0)}$. Let us call this intersection $A$ (see Figure 3) and denote it by $(\sigma_1^A, \sigma_2^A)$. Using (16) along with the fact that $A$ is on the line (19), we can derive that

$$\frac{2 + (\sigma_1^A)^2}{\gamma_1^2} = \frac{2 + (\sigma_2^A)^2}{\gamma_2^2} = 2\alpha - 1. \tag{20}$$

Now, let us consider the deviation to $(\tilde{\sigma}_1, \sigma_2^A)$ which is on the boundary of $\tilde{\Sigma}_{(1,1)}$ and $\tilde{\Sigma}_{(1,0)}$, i.e.,

$$1 - \alpha\mathcal{I}(\tilde{\sigma}_1, \sigma_2^A) = \frac{1}{2} - \alpha\mathcal{I}(\tilde{\sigma}_1). \tag{21}$$

Let

$$\eta = \frac{\gamma_1^2}{2 + \tilde{\sigma}_1{}^2}. \tag{22}$$

Plugging (20) and (22) into (21) and simplifying the equation yields

$$\eta \in \left\{ 1 - \frac{1}{4\alpha} - \frac{\sqrt{16\alpha^2 - 16\alpha + 1}}{4\alpha}, 1 - \frac{1}{4\alpha} + \frac{\sqrt{16\alpha^2 - 16\alpha + 1}}{4\alpha} \right\}. \tag{23}$$

Next we argue that we should pick the smaller (first) solution. Notice that since $\tilde{\sigma}_1 \geq \sigma_1^A$, we have $\eta = \mathcal{I}(\tilde{\sigma}_1) \leq \mathcal{I}(\sigma_1^A) = 1/(2\alpha - 1)$. In addition, Assumption 2, or equivalently (9), implies $\alpha \geq 3/2$. In this regime, we have $1 - \frac{1}{4\alpha} > \frac{1}{2\alpha-1}$. Thus, we should pick the smaller solution above.

Now, we check the condition (17). Notice that we need to find $\alpha$ such that

$$\eta + \frac{1}{2\alpha - 1} \leq \frac{1}{\alpha}. \tag{24}$$

We can verify that the right-hand side minus the left-hand side is equal to zero at $\bar{\alpha} \approx 1.88$ and positive afterward. Notice that $\eta$ and therefore the condition (24) is symmetric with respect to both platforms, and consequently, considering the other deviation to $(\sigma_1^A, \tilde{\sigma}_2)$ leads to the exact same condition on $\alpha$. This completes the proof of our claim that either both conditions hold if $\alpha \geq \bar{\alpha}$ and neither hold if $\alpha < \bar{\alpha}$. ∎

**Proof of Proposition 3**

Suppose an equilibrium $(\sigma_1^*, \sigma_2^*)$ exists in which both platforms enter the market. As we argued in Appendix C, in this case $(\sigma_1^*, \sigma_2^*) \in \tilde{\Sigma}_{(1,1)} \cap \tilde{\Sigma}_{(0,0)}$. Next, notice that for any $i \in \{1, 2\}$, the platform $i$'s utility should be nonnegative. Therefore, we have

$$\frac{1}{2} - c_i + \beta(\mathcal{I}(\sigma_1^*, \sigma_2^*) - \mathcal{I}(\sigma_{-i}^*)) \geq 0. \tag{25}$$

Notice that, since $(\sigma_1^*, \sigma_2^*) \in \tilde{\Sigma}_{(1,1)} \cap \tilde{\Sigma}_{(0,0)}$, we have $\mathcal{I}(\sigma_1^*, \sigma_2^*) = 1/\alpha$. Also, the minimum of $\mathcal{I}(\sigma_{-i}^*)$ corresponds to point $B$ for $i = 1$ and point $C$ for $i = 2$ in Figure 2b and is equal to $1/(2\alpha)$. This implies the lower bound on $\beta$.

Now, suppose $\beta \geq \bar{\beta}$. We claim the point $(\sigma_1^A, \sigma_2^A)$, defined in the proof of Lemma 6, can be an equilibrium. We just need to check when the platform's utility is nonnegative. In particular, the first platform's utility at $(\sigma_1^A, \sigma_2^A)$ is given by

$$\frac{1}{2} - c_1 + \beta\left(\mathcal{I}(\sigma_1^A, \sigma_2^A) - \mathcal{I}(\sigma_2^A)\right) = \frac{1}{2} + \frac{\beta(\alpha - 1)}{\alpha(2\alpha - 1)}, \tag{26}$$

which is nonnegative for $\beta \geq \bar{\beta}$. We can similarly write the second platform's utility at the point $A$. ∎

**Proof of Theorem 1**

We make use of the following lemmas in this proof.

**Lemma 7.** *For any $e$, the noise level equilibrium strategy, if it exists, belongs to the boundary of the set $\Sigma_{1,e}$.*

*Proof of Lemma 7:* Using Proposition 2, we know that the equilibrium noise variance belongs to $\Sigma_{1,e}$. We next argue that it cannot belong to the interior of $\Sigma_{1,e}$. To see this, suppose the contrary that $\boldsymbol{\sigma} \in \text{int}(\Sigma_1, e)$ is an equilibrium. There exists $i \in [K]$ such that by locally decreasing $\sigma_i$, we remain in the set $\Sigma_{1,e}$. We claim that platform $i$ has an incentive to decrease its noise variance. First, note that by decreasing its noise variance, we remain in the set $\Sigma_{1,e}$ and that the user still finds it optimal to share with platform $i$. However, the payment she receives from the buyer, which is given by

$$\beta\left(\mathcal{I}(\boldsymbol{\sigma}, e, a, 1) - \mathcal{I}(\boldsymbol{\sigma}, e, a, (b_i = 0, 1_{-i}))\right)$$

increases because of Lemma 4. ∎

**Lemma 8.** *For any $e$, the noise level equilibrium strategy, if it exists, belongs to $\Sigma_{\mathbf{1},e} \cap \Sigma_{\mathbf{0},e}$.*

*Proof of Lemma 8:* Using Lemma 7, we know that the equilibrium noise level, if it exists, belongs to the boundary of $\Sigma_{\mathbf{1},e}$ and $\Sigma_{\mathbf{a},e}$ for some $\mathbf{a} \in \{0,1\}^K \setminus \{\mathbf{1}\}$. We argue that it cannot belong to $\Sigma_{\mathbf{1},e} \cap \Sigma_{\mathbf{a},e}$ when $\mathbf{a} \neq \mathbf{0}$. Suppose the contrary that $\sigma \in \Sigma_{\mathbf{1},e} \cap \Sigma_{\mathbf{a},e}$ is an equilibrium where $a_i = 1$ and also $a_j = 0$. We claim that platform $i$ has an incentive to decrease its noise level. In particular, we argue that by decreasing its noise level, we go to $\text{int}(\Sigma_{\mathbf{1},e})$ and that platform $i$'s utility increases. Notice that it suffices to show that by decreasing $\sigma_i$, we remain in the set $\Sigma_{\mathbf{1},e}$. This is because by invoking Lemma 4, the payment of platform $i$ increases, and the users keep sharing with platform $i$. For $\sigma$, the indifference condition for the user implies

$$\sum_{i: \, e_i = 1} a_i \mathcal{I}_i - \alpha \mathcal{I}(\sigma, e, a, \mathbf{1}) = \sum_{i: \, e_i = 1} \mathcal{I}_i - \alpha \mathcal{I}(\sigma, e, \mathbf{1}, \mathbf{1})$$

which can be written as

$$\sum_{i: \, e_i = 1, a_i = 0} \mathcal{I}_i = \alpha \left( \mathcal{I}(\sigma, e, \mathbf{1}, \mathbf{1}) - \mathcal{I}(\sigma, e, a, \mathbf{1}) \right).$$

Now if $a_i = 1$, then by decreasing $\sigma_i$, the left-hand side does not change, and the right-hand decreases (from Lemma 4). Therefore, by decreasing $\sigma_i$, we remain in the set $\Sigma_{\mathbf{1},e}$. ∎

For a given $e$ with $e_i = 1$, platform $i$ finds it optimal to enter the market if

$$\frac{1}{2} - c_i + \beta \left( \mathcal{I}(\sigma, e, (a_i = 1, a_{-i}), \mathbf{1}) - \mathcal{I}(\sigma, e, (a_i = 0, a_{-i}), \mathbf{1}) \right) \geq 0.$$

Therefore, by letting

$$\beta \geq \bar{\beta} := \sup_{i \in [K], (e_i = 1, e_{-i}), \sigma \in \Sigma_{\mathbf{1},e} \cap \Sigma_{\mathbf{0},e}} \frac{c_i - \frac{1}{2}}{\mathcal{I}(\sigma, e, (a_i = 1, a_{-i}), \mathbf{1}) - \mathcal{I}(\sigma, e, (a_i = 0, a_{-i}), \mathbf{1})},$$

all platforms enter the market. We next argue that this term is finite. Note that, for any given value of $e_{-i}$, the fact that $\sigma \in \Sigma_{\mathbf{1},e} \cap \Sigma_{\mathbf{0},e}$ implies that the term $\mathcal{I}(\sigma, e, (a_i = 1, a_{-i}), \mathbf{1})$ in the denominator is equal to

$$\frac{|\{i : e_i = 1\}|}{2\alpha}.$$

As a result, we can rewrite $\bar{\beta}$ as

$$\bar{\beta} = \max_{i \in [K], (e_i = 1, e_{-i})} \frac{c_i - 1/2}{|\{i : e_i = 1\}|/(2\alpha) - \sup_{\sigma \in \Sigma_{\mathbf{1},e} \cap \Sigma_{\mathbf{0},e}} \mathcal{I}(\sigma, e, (a_i = 0, a_{-i}), \mathbf{1})}.$$

Notice that, for any given $\beta_{-i}$, the denominator is positive because the amount of leaked information strictly decreases when one less user shares information. Finally, $\bar{\beta}$ is finite becasue the outer maximum is taken over finitely many elements.

Similarly, for a given $e$ with $e_i = 1$, when $c_i > 1/2$ platform $i$ finds it optimal to not enter the market if

$$\frac{1}{2} - c_i + \beta \left( \mathcal{I}(\sigma, e, (a_i = 1, a_{-i}), \mathbf{1}) - \mathcal{I}(\sigma, e, (a_i = 0, a_{-i}), \mathbf{1}) \right) \leq 0.$$

Therefore, by letting

$$\beta \leq \underline{\beta} := \inf_{i \in [K], c_i > \frac{1}{2}, (e_i = 1, e_{-i}), \sigma \in \Sigma_{\mathbf{1},e}} \frac{c_i - \frac{1}{2}}{\mathcal{I}(\sigma, e, (a_i = 1, a_{-i}), \mathbf{1}) - \mathcal{I}(\sigma, e, (a_i = 0, a_{-i}), \mathbf{1})},$$

all platforms enter the market.

We next prove that assuming a set of platforms that have entered the market, for large enough $\alpha$, there exists an equilibrium. Without loss of generality, let us assume all platforms have entered the market. $\boldsymbol{\sigma} \in \Sigma_{1,1} \cap \Sigma_{0,1}$ is an equilibrium noise level if no platform has a deviation. Let us consider platform 1. For this platform, the possible deviations are to $\Sigma_{1,1} \cap \Sigma_{a,1}$ where $a_1 = 0$ and $\boldsymbol{a}_{-1}$ can be any vector in $\{0, 1\}^{K-1}$. We next argue that $\boldsymbol{\sigma} \in \Sigma_{1,1} \cap \Sigma_{0,1}$ for which

$$\frac{\gamma_i}{2 + \sigma_i^2} = t \text{ for all } i \in [K]$$

is a noise level equilibrium for large enough $\alpha$. Similar to the analysis with $K = 2$ platforms, we let

$$\mathcal{I}(\sigma_i : i \in S)$$

for any $S \subseteq [K]$ denote the leaked information about $\theta$ when the data of the user is shared with the buyer through platforms in the set $S$ and platform $i \in S$ is using noise variance $\sigma_i^2$.

Before proceeding with the proof, we first find a closed-form expression for the leaked information.

**Lemma 9.**

- *Suppose $\frac{\gamma_i}{2+\sigma_i^2} = t$ for all $i \in [K]$. The leaked information about $\theta$ when the data of all platforms is shared with the buyer becomes*

$$\frac{Kt}{1 + (K-1)t}.$$

- *Suppose $\frac{\gamma_i}{2+\sigma_i^2} = t$ for all $i \in [K] \setminus \{1\}$ and $\frac{\gamma_1}{2+\sigma_1^2} = t'$. The leaked information about $\theta$ when the data of all platforms is shared with the buyer becomes*

$$1 + \frac{t-1}{1 + t' + (K-2)t}.$$

*Proof:* Using Lemma 1, the leaked information is

$$(\gamma_1, \ldots, \gamma_K) \begin{pmatrix} \gamma_1^2/t & \gamma_1\gamma_2 \cdots & \gamma_1\gamma_K \\ \gamma_2\gamma_1 & \gamma_2^2/t \cdots & \gamma_2\gamma_K \\ \vdots & & \\ \gamma_K\gamma_1 & \gamma_K\gamma_2 \cdots & \gamma_K^2/t \end{pmatrix}^{-1} (\gamma_1, \ldots, \gamma_K)^T$$

$$\overset{(a)}{=} \boldsymbol{\gamma}^T \left( \boldsymbol{\gamma}\boldsymbol{\gamma}^T + (\frac{1}{t} - 1)\text{diag}(\gamma_1^2, \ldots, \gamma_K^2) \right)^{-1} \boldsymbol{\gamma}$$

$$\overset{(a)}{=} \boldsymbol{\gamma}^T \left( \frac{1}{\frac{1}{t} - 1}\text{diag}(1/\gamma_1^2, \ldots, 1/\gamma_K^2) - \frac{\frac{1}{\frac{1}{t}-1}\text{diag}(1/\gamma_1^2, \ldots, 1/\gamma_K^2)\boldsymbol{\gamma}^T\boldsymbol{\gamma}\frac{1}{\frac{1}{t}-1}\text{diag}(1/\gamma_1^2, \ldots, 1/\gamma_K^2)}{1 + \boldsymbol{\gamma}^T\frac{1}{\frac{1}{t}-1}\text{diag}(\gamma_1^2, \ldots, \gamma_K^2)\boldsymbol{\gamma}} \right) \boldsymbol{\gamma}$$

$$= \frac{Kt}{1 + (K-1)t},$$

where in (a), we used $\boldsymbol{\gamma} = (\gamma_1, \ldots, \gamma_K)^T$ and in (b), we used the Sherman–Morrison formula for the inverse of a rank-one perturbation of a matrix. This completes the proof of the first part. The second part follows from a similar argument. ∎

Let us consider the deviation of platform 1 so that the corresponding user equilibrium is $(a_1 = 1, \boldsymbol{a}_{-i})$ with $\boldsymbol{a}_{-i}$ having $i - 1$ ones and $K - i$ zeros. Notice that if such deviation is not possible by increasing $\sigma_1$, then this is not a profitable deviation for platform 1, and we now verify that even if such deviation is possible, it is not profitable for platform 1. Given that $\boldsymbol{\sigma} \in \Sigma_{\mathbf{1,1}} \cap \Sigma_{\mathbf{0,1}}$, we have that the user is indifferent between sharing with all platforms and sharing with none of them. Therefore, we have

$$\sum_{i=1}^{K} \mathcal{I}_i - \alpha \mathcal{I}(\sigma_1, \ldots, \sigma_K) = 0$$

which gives us

$$\mathcal{I}(\sigma_1, \ldots, \sigma_K) = \frac{K}{2\alpha}.$$

Using Lemma 9, the above equality gives us

$$t = \frac{1}{1 + 2\alpha - K}.$$

Given the above deviation to $(\tilde{\sigma}_1, \ldots, \sigma_K)$, the user indifference condition gives us

$$\frac{K}{2} - \alpha \mathcal{I}(\tilde{\sigma}_1, \ldots, \sigma_K) = \frac{i}{2} - \alpha \mathcal{I}(\tilde{\sigma}_1, \ldots, \sigma_i).$$

Substituting the closed-form expressions of Lemma 9 in the above equality gives us

$$t' = \frac{2 - 4\alpha + K - i - \sqrt{16\alpha^2 + (K - i)^2 - 8\alpha K}}{2 + 4\alpha - 2K},$$

where $t' = \frac{\gamma_1^2}{2 + \tilde{\sigma}_1^2}$.

Now, for this deviation to be not profitable, we must have

$$\mathcal{I}(\sigma_1, \ldots, \sigma_K) - \mathcal{I}(\sigma_2, \ldots, \sigma_K) \geq \mathcal{I}(\tilde{\sigma}_1, \ldots, \sigma_i) - \mathcal{I}(\sigma_2, \ldots, \sigma_i).$$

Again, using the closed-form expressions of Lemma 9 and writing $t'$ and $t$ in terms of $\alpha$, the above inequality becomes equivalent to

$$-1 + \frac{1 - K}{2\alpha - 1} + \frac{i - 1}{2\alpha - 1 + i - K} + \frac{K}{2\alpha} + \frac{2K - 4\alpha}{K - i + \sqrt{16\alpha^2 + (K - i)^2 - 8\alpha K}} \geq 0,$$

which we prove next. We can write

$$\frac{3K - i}{4\alpha} + \frac{1 - K}{2\alpha - 1} + \frac{i - 1}{2\alpha - 1 + i - K} - 1 + \frac{\sqrt{16\alpha^2 + (K - i)^2 - 8\alpha K}}{4\alpha}$$

$$\stackrel{(a)}{\geq} \frac{3K - i}{4\alpha} + \frac{1 - K}{2\alpha - 1} + \frac{i - 1}{2\alpha - 1 + i - K} - \frac{K}{2\alpha \left( 1 + \sqrt{1 - \frac{K}{2\alpha}} \right)}$$

$$\geq \frac{3K - 1}{4\alpha} - \frac{K - 1}{2\alpha} - \frac{K}{2\alpha \left( 1 + \sqrt{1 - \frac{K}{2\alpha}} \right)} \stackrel{(b)}{\geq} 0$$

where (a) follows by noting that the last term is increasing in $i$ and (b) holds for

$$\alpha \geq \bar{\alpha} := \frac{(K+1)^2}{8}.$$

Notice that for any other set $S \subseteq [K]$ of the platforms that have entered the market, the above bound on $\alpha$ guarantees the existence of platform equilibrium strategy as $|S| \leq K$. This completes the proof. ∎

**Proof of Proposition 6**

The user utility from sharing if the platform enters the market is

$$\mathcal{I}_1 - \alpha \mathcal{I}(\sigma_1) = \frac{1}{2} - \alpha \frac{\gamma_1^2}{2 + \sigma_1^2}.$$

The above utility is always positive if $\alpha \leq 1/\gamma_1^2$. Therefore, the user always shares their data. This also implies that the platform chooses $\sigma_1 = 0$ to maximize the payment received from the buyer. The platform's utility then becomes

$$\mathcal{I}_1 - c_1 + \beta \mathcal{I}(0) = \frac{1}{2} - c_1 + \beta \frac{\gamma_1^2}{2}.$$

Therefore, the platform enters the market if and only if the above quantity is positive.

If $\alpha > 1/\gamma_1^2$, then the user's utility is not always positive. In this case, if the platform enters the market, it chooses $\sigma_1$ so that the user utility is zero, i.e.,

$$\mathcal{I}_i - \alpha \mathcal{I}(\sigma_1) = \frac{1}{2} - \alpha \frac{\gamma_1^2}{2 + \sigma_1^2} = 0.$$

The above equality gives us $\sigma_1 = \sqrt{2\left(\alpha\gamma_1^2 - 1\right)}$. The platform's utility then becomes

$$\mathcal{I}_1 - c_1 + \beta \mathcal{I}(\sigma_1) = \frac{1}{2} - c_1 + \beta \frac{1}{2\alpha}.$$

Therefore, the platform enters the market if and only if the above quantity is positive. ∎

**Proof of Corollary 1**

When all platforms enter the market, as we have established in Lemma 8, the equilibrium noise variance is such that the user is indifferent between sharing with all platforms and not sharing at all. This means that

$$\sum_{i=1}^{K} \mathcal{I}_i - \alpha \mathcal{I}(\sigma_1, \dots, \sigma_K) = 0. \tag{27}$$

Moreover, the payments cancel out in the utilitarian welfare and it becomes

$$2 \sum_{i=1}^{K} \mathcal{I}_i + (\beta - \alpha) \mathcal{I}(\sigma_1, \dots, \sigma_K) - \sum_{i=1}^{K} c_i = K + (\beta - \alpha) \frac{K}{2\alpha} - \sum_{i=1}^{K} c_i,$$

where the above equality follows from $\mathcal{I}_i = \frac{1}{2}$ for all $i \in [K]$ and (27).

The argument is similar to the above one when only low-cost platforms enter the market. ∎

**Proof of Proposition 7**

If we only have low-cost platforms, then all platforms enter the market as their utility of platform $i \in [K]$ is given by

$$\mathcal{I}_i - c_i + p_i = \frac{1}{2} - c_i + p_i \geq 0,$$

which is always positive. Notice that if we ban data sharing, then the user shares with all platforms (because there is no privacy loss), and its utility becomes

$$\sum_{i=1}^{K} \mathcal{I}_i = \frac{K}{2}.$$

Clearly, there is no uniform minimum privacy mandate that can make the user better off because banning data sharing maximizes the user gain from services and minimizes the privacy loss.

Now, let us prove the second part of this proposition. By banning data sharing, only low-cost platforms enter the market, and the user shares data with all of them and obtains the following utility:

$$\frac{1}{2}( \text{ number of low-cost platforms}).$$

Now consider a uniform minimum privacy mandate $\bar{\sigma}$ such that

$$\frac{1}{2}K - \alpha \mathcal{I}(\underbrace{\bar{\sigma}, \ldots, \bar{\sigma}}_{K \text{ many}}) > \frac{1}{2}( \text{ number of low-cost platforms}).$$

For any $\alpha$, as $\bar{\sigma}$ increases, the leaked information goes to zero, and therefore, so long as there exists at least one high-cost platform, a $\bar{\sigma}$ for which the above inequality holds exist. We next argue that all platforms enter the market for large enough $\beta$. To see this, we need to have

$$\mathcal{I}_i - c_i + \beta \left( \mathcal{I}(\underbrace{\bar{\sigma}, \ldots, \bar{\sigma}}_{K \text{ many}}) - \mathcal{I}(\underbrace{\bar{\sigma}, \ldots, \bar{\sigma}}_{K-1 \text{ many}}) \right) \geq 0$$

for all $i \in [K]$. Therefore, by letting $\beta \geq \tilde{\beta}$ where

$$\tilde{\beta} = \max \left\{ \bar{\beta}, \frac{c_i - \frac{1}{2}}{\mathcal{I}(\underbrace{\bar{\sigma}, \ldots, \bar{\sigma}}_{K \text{ many}}) - \mathcal{I}(\underbrace{\bar{\sigma}, \ldots, \bar{\sigma}}_{K-1 \text{ many}})} \right\}$$

and $\bar{\beta}$ is specified in Theorem 1 all platforms enter the market. ∎

**Proof Proposition 8**

Similar to the proof of Proposition 7 and Theorem 1, we choose $\bar{\alpha}$ and $\hat{\beta}$ large enough so that an equilibrium exists and all platforms enter the market. Now suppose $\bar{\sigma}$ is the optimal uniform minimum privacy mandate so that all platforms enter the market, the user utility is

$$\frac{1}{2}K - \alpha \mathcal{I}(\underbrace{\bar{\sigma}, \ldots, \bar{\sigma}}_{K \text{ many}}),$$

and that the utility of platform $i$ is

$$\mathcal{I}_i - c_i + \beta\Big(\mathcal{I}(\underbrace{\bar\sigma,\ldots,\bar\sigma}_{K\text{ many}}) - \mathcal{I}(\underbrace{\bar\sigma,\ldots,\bar\sigma}_{K-1\text{ many}})\Big) \geq 0.$$

Consider a regulation that bans data sharing for low-cost platforms and uses the same minimum privacy mandate for high-value platforms. We first prove that all platforms still enter the market. First, notice that the low-cost platforms enter the market as they obtain a positive utility. Letting $h$ be the number of high-cost platforms, the utility of a high-cost platform $i$ becomes

$$\mathcal{I}_i - c_i + \beta\Big(\mathcal{I}(\underbrace{\bar\sigma,\ldots,\bar\sigma}_{h\text{ many}}) - \mathcal{I}(\underbrace{\bar\sigma,\ldots,\bar\sigma}_{h-1\text{ many}})\Big) \geq \mathcal{I}_i - c_i + \beta\Big(\mathcal{I}(\underbrace{\bar\sigma,\ldots,\bar\sigma}_{K\text{ many}}) - \mathcal{I}(\underbrace{\bar\sigma,\ldots,\bar\sigma}_{K-1\text{ many}})\Big) \geq 0,$$

where the first inequality follows from the submodularity of the leaked information in actions, established in Lemma 3. The utility of the user also increases because we have

$$\frac{1}{2}K - \alpha\mathcal{I}(\underbrace{\bar\sigma,\ldots,\bar\sigma}_{h\text{ many}}) \geq \frac{1}{2}K - \alpha\mathcal{I}(\underbrace{\bar\sigma,\ldots,\bar\sigma}_{K\text{ many}}),$$

where the inequality follows from the monotonicity of the leaked information, established in Lemma 2. ■

