# OpenReview forum: "On Three-Layer Data Markets"
_ICML.cc/2024/Workshop/Agentic_Markets — Agentic Markets @ ICML'24 Oral_

### Official Review · Reviewer_9a8T · 2024-06-06
**Review for submission 14**

**Rating:** 7
**Confidence:** 4

**Review:**

**Summary:** The paper studies the problem of selling data to third parties by platforms that serve many users. Specifically, platforms choose whether to operate in a competitive market or not in which they offer services to users. They also decide what level of data privacy to offer to users and at what price to sell the collected data to third party buyers. The paper finds the equilibrium of this 3-party market and argues about user, buyer and platform welfare and how data-related policies, e.g., such as GDPR, may have various effects on the welfare of these 3 types of participants.

**Evaluation:** The paper studies an interesting problem and does a very good job in motivating it. I also appreciate how the authors compressed the gist of their paper in 4 pages (I only skimmed through the appendix). The approach is rigorous and the model seems rich and complex enough to support interesting findings. On the flipside, I believe that many of the assumptions and conclusions of the model need better justification - so, I consider the current status of the paper as an intermediate draft before the paper is ready for proper publication. However, the paper is well-written, falls within the scope of the workshop and provides insights (even if incomplete) into a topic can prompt interesting discussions, so I am in favour of acceptance.

**Comments to authors:**
- Offering concrete examples of high-cost firms (close to the bottom of page 2) will improve the exposition in that part. Are travel/event online booking agencies a good example? But again, are there firms that make profit *only* from selling data to third parties? Shall one think of websites like free online newspapers or blogs here (were users pay nothing, but they still collect data about their interests)?
- The sequence of actions needs to be better justified, in particular, why pricing by platforms comes third (late). Maybe this is not important (and I misunderstood) and, so, considering alternative orderings (in which pricing comes early) is easier and can be used to showcase robustness of results (Nash equilibrium outcomes) which would strengthen the paper.
- Similarly justifying why privacy level is modelled as noise should be better justified (and potentially compared to deterministic levels) for robustness. I appreciate the two references, but aren't there papers that take the latter approach, i.e., that of deterministic levels?
- Can you clarify the following: "and when the user is privacy-conscious and requires a minimum non-zero level of noise for data sharing, the user utility is equal to zero at equilibrium". I thought that users prefer a "max" level of sharing, i.e., users prefer less noise.
- The next sentence is also confusing: "Here, both the user and the data buyer utilities are zero at equilibrium, indicating that having both platforms in the market to compete benefits the data buyers but not the user." But, right before, the paper indicated that buyers had positive utility for high \beta.
- When all platforms are in the market, does the buyer buy only from one or from (potentially) more than one. I refer to the last paragraph of the Intro starting with "We demonstrate that...". If all platforms enter the market but the buyer deterministically only buys from one, will not this result back to having a concentrated market?
- I also struggled to follow the 3 insights described in the "policy and regulation" paragraph. The text seems to imply a contradiction between the first and second insight, but both suggest that a uniform minimum (again, why a minimum and not a max?) privacy mandate improves user utility. The text that follows confuses things even more. I think that the effects of different parameters have not been disentangled here and a more careful approach to drawing conclusions (regarding the effects of policies and regulations) is needed.
- Since different policies seem to have different effects on different types of participants would it make to also gauge a social welfare metric to evaluate the outcomes of different policies?

---

### Official Review · Reviewer_poJJ · 2024-06-18
**3. #14 On Three-Layer Data Markets**

**Rating:** 9
**Confidence:** 3

**Review:**

The paper presents a theoretical model for analyzing the Nash equilibrium between users, buyers of data, and platforms, showcasing that there is a middle ground that is generally not in the user's benefit under policies like GDPR that impose restrictions on competitiveness in the marketplace generally. It argues for a better implementation of data privacy regulation, using prior literature from economics as well as platform business models to inform their suggested policies, and then offering theoretical justification for their conclusions. They generalize their findings to a larger number of platforms in this marketplace to drive home the point about competitiveness and how it does not increase user utility while increasing others' utility and value from data. The paper is a well-constructed set of theories with meaningful policy implications that are argued for in a rigorous framework with potential to influence regulatory policymaking around user data management on platforms. The main text of the paper is limited to a brief motivation, literature review, and introduction of the problem setting. The majority of their claimed results are relegated to the supplementary materials. The paper is non-anonymized which is a fundamental problem with their submission. Nevertheless, I have attempted to offer an objective review of the text.

I have not validated the proofs presented within the supplementary materials.

Recommend an Oral Presentation (4.5/5) with medium confidence.